# TokenSwift: Lossless Acceleration of Ultra Long Sequence Generation

Tong Wu [* 1]   Junzhe Shen [* 1 2]   Zixia Jia [1]   Yuxuan Wang [1]   Zilong Zheng [1]

## Abstract

Generating ultra-long sequences with large language models (LLMs) has become increasingly crucial but remains a highly time-intensive task, particularly for sequences **up to 100K tokens**. While traditional speculative decoding methods exist, simply extending their generation limits fails to accelerate the process and can be detrimental. Through an in-depth analysis, we identify three major challenges hindering efficient generation: frequent model reloading, dynamic key-value (KV) management and repetitive generation. To address these issues, we introduce **TOKENSWIFT**, a novel framework designed to substantially accelerate the generation process of ultra-long sequences while maintaining the target model's inherent quality. Experimental results demonstrate that TOKENSWIFT achieves over $3\times$ speedup across models of varying scales (1.5B, 7B, 8B, 14B) and architectures (MHA, GQA). This acceleration translates to hours of time savings for ultra-long sequence generation, establishing TOKENSWIFT as a scalable and effective solution at unprecedented lengths.

## 1. Introduction

Recent advances in large language models (LLMs), amplified by their long context capacities (Wu et al., 2024; Ding et al., 2024), have demonstrated remarkable proficiency in intricate reasoning (Jaech et al., 2024; Guo et al., 2025), agentic thinking (Shinn et al., 2023; Yao et al., 2023; Li et al., 2024a), and creative writing (Wang et al., 2023; Mikhaylovskiy, 2023), *etc*. These advancements necessitate the ability to generate lengthy sequences, *e.g.*, o1-like (Jaech et al., 2024) reasoning tends to generate protracted chain-of-thought trajectories before reaching final conclusions.

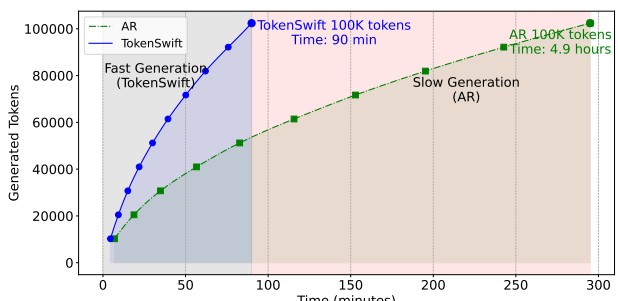

*Figure 1.* Comparison of the time taken to generate 100K tokens using autoregressive (AR) and TOKENSWIFT with prefix length of 4096 on `LLaMA3.1-8b`. As seen, TOKENSWIFT accelerates the AR process from nearly 5 hours to just 90 minutes.

However, a critical challenge impeding the practical deployment of such applications is the extensive time required to produce ultra-long sequences. For instance, generating 100K tokens with LLaMA3.1-8B can take approximately five hours (Figure 1), a duration that is impractically long for the development of sophisticated applications, let alone recent gigantic models such as LLaMA3.1-405B (AI@Meta, 2024) and DeepSeek-671B (Liu et al., 2024a). Addressing this bottleneck is essential for harnessing the full potential of LLMs in real-world scenarios.

A straightforward solution is to take advantage of recent success in speculative decoding (SD) (Leviathan et al., 2023; Chen et al., 2023), which employs a *draft-then-verify* strategy to expedite generation while preserving *lossless* accuracy; see Appendix A and Section 5.1 for detailed background and relevant literature. However, these methods are generally tailored for generating short sequences, *e.g.*, TriForce (Sun et al., 2024a) and MagicDec (Chen et al., 2024a) are limited to generating 256 and 64 tokens, respectively. Directly extending their generation length to 100K tokens would inevitably encounter failures due to KV cache budget constraints. Furthermore, even when applied to optimized KV cache architectures such as Group Query Attention (GQA), these methods yield only marginal acceleration gains for short-sequence generation, as evidenced in Tables 1 and 3. This observation leads to a pivotal research question:

*Is it possible to achieve model-agnostic **lossless** accelerations, akin to those seen in short-sequence SDs, for generating **ultra-long** sequences, with **minimal** training overhead?*

[*]Equal contribution [1]State Key Laboratory of General Artificial Intelligence, BIGAI, Beijing, China [2]LUMIA Lab, Shanghai Jiao Tong University. Correspondence to: Zilong Zheng <zlzheng@bigai.ai>.

*Proceedings of the $42^{nd}$ International Conference on Machine Learning*, Vancouver, Canada. PMLR 267, 2025. Copyright 2025 by the author(s).

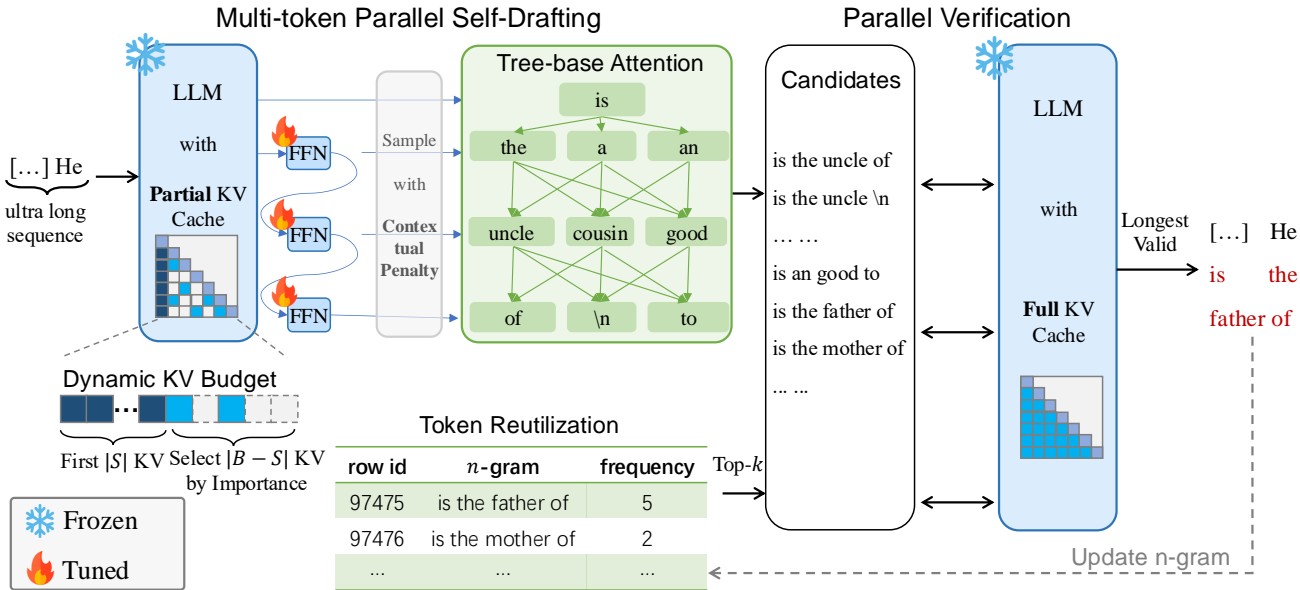

*Figure 2.* **Illustration of TOKENSWIFT Framework.** First, target model (LLM) with partial KV cache and three linear layers outputs 4 logits in a single forward pass. Tree-based attention is then applied to construct candidate tokens. Secondly, top-$k$ candidate 4-grams are retrieved accordingly. These candidates compose draft tokens, which are fed into the LLM with full KV cache to generate target tokens. The verification is performed by checking if draft tokens match exactly with target tokens (Algorithm 1). Finally, we randomly select one of the longest valid draft tokens, and update $n$-gram table and KV cache accordingly.

To answer this question, we conduct an in-depth analysis (§2) and identify three key challenges: **(1)** *frequent model reloading*: frequently reloading model for each token generation introduces a significant delay, primarily due to memory access times rather than computation. **(2)** *Prolonged Growing of KV Cache*, the dynamic management of key-value (KV) pairs, which grow with the sequence length, adds complexity in maintaining model efficiency. **(3)** *repetitive content generation*, the issue of repetitive generation becomes more pronounced as the sequence length increases, leading to degraded output quality.

Building on these insights, we introduce our framework TO-KENSWIFT, which utilizes $n$-gram retrieval and dynamic KV cache updates to accelerate ultra-long sequence generation. Specifically, we employ *multi-token generation* and *token reutilization* to enable the LLM (*i.e.* target model) to draft multiple tokens in a single forward pass, alleviating the first challenge of frequent model reloading (§3.2). As the generation progresses, we *dynamically update* the partial KV cache at each iteration, reducing the KV cache loading time (§3.3). Finally, to mitigate the issue of repetitive outputs, we apply *contextual penalty* to constrain the generation process, ensuring the diversity of output (§3.4).

In §4, we conduct extensive experiments to evaluate TO-KENSWIFT across different model scales and architectures. In summary, we highlight our advantages as:

1. To the best of our knowledge, TOKENSWIFT is the **first** to accelerate ultra-long sequence generation up to 100K with lossless accuracy of target LLMs, while demonstrat-ing significant superiority over enhanced baselines.

2. TOKENSWIFT consistently achieves over $3\times$ speedup compared to AR across varying prefix lengths, model architectures, and model scales in generating 100K tokens, reducing the AR process from nearly 5 hours to 90 minutes on LLaMA3.1-8b.

3. TOKENSWIFT achieves progressively higher speedup compared to AR as the generation length increases, while enhancing diversity in ultra-long sequence generation (as measured by *Distinct-n* (Li et al., 2016)).

## 2. Challenges

Accelerating long sequence generation is nevertheless a non-trivial task, even built upon prior success in speculative decoding (SD). In this section, we identify critical challenges encountered in accelerating ultra-long sequence generation.

**Challenge I: Frequent Model Reloading**  One fundamental speed obstacle lies in the autoregressive (AR) generation scheme of LLM. For each token, the entire model must be loaded from GPU's storage unit to the computing unit (Yuan et al., 2024), which takes significantly more time than the relatively small amount of computation performed (as shown in Table 2). Consequently, the primary bottleneck in generation stems from I/O memory access rather than computation.

▷ *When generating ultra-long sequence, such as 100K tokens, the GPU must reload the model weights over 100,000 times. This repetitive process poses the challenge: How can we reduce the frequency of model reloading?*

*Table 1.* Experimental results of TriForce (Sun et al., 2024a) and MagicDec (Chen et al., 2024a) with default parameters on `LLaMA3.1-8b`. The Batch Size of MagicDec is set to 1. The results on A100-80G and PG-19 using identical hyperparameters. The only difference is that their original experiments were conducted on `YaRN-LLaMA2-7b-128k` (MHA), while ours used `LLaMA3.1-8b` (GQA).

| Method | Gen. Len. | Draft Form | Speed Up |
|---|---|---|---|
| **TriForce** | 256 | Standalone Draft | 1.02 |
| **MagicDec** | 64 | Self-Speculation | 1.20 |
| | | Standalone Draft | 1.06 |

*Table 2.* Taking NVIDIA A100 80G and `LLaMA3.1-8b` as example, *MAX* refers to the scenario with a maximum context window 128K. The calculation method is from Yuan et al. (2024).

| MEMORY | COMPUTATION |
|---|---|
| *Bandwidth*: 2.04e12 B/s | *BF16*: 312e12 FLOPS |
| *Model Weights*: 15.0 GB | *MAX Operations*: 83.9 GB |
| *Loading Time*: **7.4** ms | *MAX Computing Time*: **0.3** ms |

**Challenge II: Prolonged Growing of KV Cache**   Previous studies, such as TriForce (Sun et al., 2024a) and MagicDec (Chen et al., 2024a) have demonstrated that, a small KV cache budget can be used during the drafting phase to reduce the time increase caused by the loading enormous KV cache. While their one-time compression strategy at the prefill stage can handle scenarios with long prefixes and short outputs, it fails to address cases involving ultra-long outputs, as the growing size of KV cache would far exceed the allocated length budget.

▷ *To dynamically manage partial KV cache within limited budget during ultra-long sequence generation, the challenge lies in determining when and how to dynamically update the KV cache.*

**Challenge III: Repetitive Content Generation**   The degeneration of AR in text generation tasks — characterized by output text that is bland, incoherent, or gets stuck in repetitive loops — is a widely studied challenge (Holtzman et al., 2020; Nguyen et al., 2024; Hewitt et al., 2022). When generating sequences of considerable length, *e.g.*, 100K, the model tends to produce repetitive sentences (Figure 5).

▷ *Since our objective is lossless acceleration and repetition is an inherent problem in LLMs, eliminating this issue is not our focus. However, it is still essential and challenging to mitigate repetition patterns in ultra-long sequences.*

## 3. TOKENSWIFT

To achieve **lossless acceleration in generating ultra-long sequences**, we propose tailored solutions for each challenge inherent to this process. These solutions are seamlessly integrated into a unified framework, *i.e.* TOKENSWIFT.

### 3.1. Overview

The overall framework is depicted in Figure 2. TOKENSWIFT generate a sequence of draft tokens with self-drafting, which are then passed to the target (full) model for validation using a tree-based attention mechanism (See Appendix E for more tree-based attention details). This process ensures that the final generated output aligns with the target model's predictions, effectively achieving lossless acceleration.

TOKENSWIFT is lightweight because the draft model is the target model itself with a partial KV cache. This eliminates the need to train a separate draft LLM; instead, only $\gamma$ linear layers need to be trained, where $\gamma + 1$[1] represents the number of logits predicted in a single forward pass. In addition, during the verification process, once we obtain the target tokens from the target model with full KV cache, we directly compare draft tokens with target tokens sequentially to ensure that the process is lossless (He et al., 2024).

### 3.2. Multi-token Generation and Token Reutilization

**Multi-token Self-Drafting**   Inspired by Medusa (Cai et al., 2024), we enable the LLM to generate multiple draft tokens in a single forward pass by incorporating $\gamma$ additional linear layers. However, we empirically note that **the additional linear layers should not be independent of each other**. Specifically, we propose the following structure:

$$h_1 = f_1(h_0) + h_0, \quad h_2 = f_2(h_1) + h_1, \quad h_3 = f_3(h_2) + h_2,$$
$$l_0, l_1, l_2, l_3 = g(h_0), g(h_1), g(h_2), g(h_3), \quad (1)$$

where $h_0$ denotes the last hidden state of LLM, $f_i(\cdot)$ represents the $i$-th linear layer, $h_i$ refers to the $i$-th hidden representation, $g(\cdot)$ represents the LM Head of target model, and $l_i$ denotes output logits. This structure aligns more closely with the AR nature of the model. Moreover, this adjustment incurs no additional computational cost.

**Token Reutilization**   Given the relatively low acceptance rate of using linear layers to generate draft tokens, we propose a method named **token reutilization** to further reduce the frequency of model reloads. The idea behind token reutilization is that some phrases could appear frequently, and they are likely to reappear in subsequent generations.

Specifically, we maintain a set of tuples $\{(\mathcal{G}, \mathcal{F})\}$, where $\mathcal{G} = \{x_{i+1}, ..., x_{i+n}\}$ represents an $n$-gram and $\mathcal{F}$ denotes its corresponding frequency $\mathcal{F}$ within the generated token sequence $S = \{x_0, x_1, ..., x_{t-1}\}$ by time step $t$ ($t \geq n$). After obtaining $\{p_0, ..., p_3\}$ as described in §3.4, we retrieve the top-$k$ most frequent $n$-grams beginning with token $\arg\max p_0$ to serve as additional draft tokens.

Although this method can be applied to tasks with long

---

[1] The target model itself can also predict one logit, making the total number of logits $\gamma + 1$. We take $\gamma = 3$.

prefixes, its efficacy is constrained by the limited decoding steps, which reduces the opportunities for accepting $n$-gram candidates. Additionally, since the long prefix text is not generated by the LLM itself, a distributional discrepancy exists between the generated text and the authentic text (Mitchell et al., 2023). As a result, this method is particularly suitable for generating ultra-long sequences.

### 3.3. Dynamic KV Cache Management

**Dynamic KV Cache Updates**   Building upon the findings of Xiao et al. (2024), we preserve the initial $|S|$ KV pairs within the cache during the drafting process, while progressively evicting less important KV pairs. Specifically, we enforce a fixed budget size $|B|$, ensuring that the KV cache at any given time can be represented as:

$$\mathbf{KV} = \{(\mathbf{K}_0, \mathbf{V}_0), ..., (\mathbf{K}_{|S|}, \mathbf{V}_{|S|}), (\mathbf{K}_{|S|+1}, \mathbf{V}_{|S|+1}), ..., (\mathbf{K}_{|B|-1}, \mathbf{V}_{|B|-1})\},$$

where the first $|S|$ pairs remain fixed, and the pairs from position $|S|$ to $|B|-1$ are ordered by decreasing importance. As new tokens are generated, less important KV pairs are gradually replaced, starting from the least important ones at position $|B|-1$ and moving towards position $|S|$. Once replacements extend beyond the $|S|$ position, we recalculate the *importance scores* of all preceding tokens and select the most relevant $|B|-|S|$ pairs to reconstruct the cache. This process consistently preserves the critical information required for ultra-long sequence generation.

**Importance Score of KV pairs**   We rank the KV pairs based on the *importance scores* derived from the dot product between the query ($\mathbf{Q}$) and key ($\mathbf{K}$), *i.e.* $\mathbf{QK}^T$.

In the case of Group Query Attention (GQA), since each $\mathbf{K}$ corresponds to a group of $\mathcal{Q} = \{\mathbf{Q}_0, ..., \mathbf{Q}_{g-1}\}$, direct dot-product computation is not feasible. Unlike methods such as SnapKV (Li et al., 2024c), we do not replicate the $\mathbf{K}$. Instead, we partition the $\mathcal{Q}$, as shown in Equation (2):

$$\text{importance score}_i = \sum_{j=i \cdot g}^{((i+1) \cdot g)-1} \mathbf{Q}_j \cdot \mathbf{K}_i, \qquad (2)$$

where for position $i$, $\mathbf{Q}_j$ in the group $\mathcal{Q}_i$ are dot-product with the same $\mathbf{K}_i$, and their results are aggregated to obtain the final *importance score*. This approach enhances memory saving while preserving the quality of the attention mechanism, ensuring that each query is effectively utilized without introducing unnecessary redundancy.

### 3.4. Contextual Penalty and Random N-gram Selection

**Contextual Penalty**   To mitigate repetition in generated text, we have explored various sampling strategies. However, with the significantly larger sequence length, the likelihood of repetition increases significantly (§2). As a result, we decided to apply an additional penalty to the generated tokens to further mitigate repetition.

---

**Algorithm 1** TOKENSWIFT

**Require:** Prompt $\mathbf{p}$, target model $M$, decoding tree $T$, $n$-gram candidate number $k$; max budget size $|B|$ of partial cache, cache initial size $|S|$.

1: Prefill target model with KV cache $C_{full} \leftarrow Prefill_M(\mathbf{p})$, s.t. $|C_{full}| = \text{len}(\mathbf{p})$;
2: Prefill partial KV cache $C_p \leftarrow \{C_{full}[0 : |S|], \text{Top-K}_{|B|-|S|}(C_{full})\}$ *w.r.t* Equation (2), where $|C_p| = |B|$;
3: $st \leftarrow 0, e \leftarrow \text{len}(\mathbf{p})$.
4: **while** $st \leqslant$ target length **do**
5:     **if** $(|C_{full}| - e) > |B| - |S|$ **then**
6:       **Dynamic KV Cache Update:** $C_p \leftarrow \{C_{full}[: |S|], \text{Top-K}_{|B|-|S|}(C_{full})\}, e \leftarrow |C_{full}|$.
7:     **end if**
8:     **Multi-token Parallel Generation:** Get penalized probability $p_{\leqslant 3}$[2] with partial cache $C_p$.
9:     **Tree-based Attention:** Construct $g$ groups of candidate draft tokens $\{x^i_{\leqslant 3}\}^g_{i=1}$ using decoding tree $T$ and $p_{\leqslant 3}$.
10:     **Token Reutilization:** Select $k$ n-gram candidates $\{a^i_{\leqslant 3}\}^k_{i=1}$ with highest frequency, where $a_0 = \arg\max p_0$ (§3.2).
11:     **Parallel Verification:** Let draft tokens $\{d^i\}^{k+g}_{i=1} := \{a^i_{\leqslant 3}\}^k_{i=1} \cup \{x^i_{\leqslant 3}\}^g_{i=1}$, and send $\{d\}$ to $M$ to get penalized verification probabilities $\{q^i\}^{k+g}_{i=1}$.
12:     Sample target tokens $\{y^i\}^{k+g}_{i=1} \sim \{q^i\}^{k+g}_{i=1}$.
13:     Random select the longest accepted length of draft tokens $d^j_{\leqslant m} \in \{d_{\leqslant m} | d_{\leqslant m} = y^i_{\leqslant m}\}^{k+g}_{i=1}$ by exactly match.
14:     Let $st \leftarrow st + \text{len}(y^i)$; **yield:** $y^i$
15:     Evict $C_p$ to ensure the size of $C_p$ is $|B|$ and update $C_{full}$.
16: **end while**

---

The penalized sampling approach proposed in (Keskar et al., 2019) suggests applying a penalty to all generated tokens. However, when generating ultra-long sequences, the set of generated tokens may cover nearly all common words, which limits the ability to sample appropriate tokens. Therefore, we propose an improvement to this method.

Specifically, we introduce a fixed *penalty window $W$* and apply *penalty value $\theta$* to the most recent $W$ tokens, denoted as $\mathbb{W}$, generated up to the current position, as illustrated in Equation (3):

$$p_i = \frac{\exp\left(l_i / (t \cdot I(l_i))\right)}{\sum_j \exp\left(l_j / (t \cdot I(l_j))\right)}, \qquad (3)$$
$$I(l) = \theta \text{ if } l \in \mathbb{W} \text{ else } 1.0, \quad \theta \in (1, \infty),$$

where $t$ denotes temperature, $l_i$ and $p_i$ represent the logit and probability of $i$-th token. This adjustment aims to maintain diversity while still mitigating repetitive generation.

**Random $n$-gram Selection**   In our experiments, we observe that the draft tokens provided to the target model for

---

[2]The subscript $\leqslant 3$ here denotes a tuple with indices $0, 1, 2, 3$. The notation will be used similarly hereafter.

parallel validation often yield multiple valid groups. Building on this observation, we randomly select one valid $n$-gram to serve as the final output. By leveraging the fact that multiple valid $n$-grams emerge during verification, we ensure that the final output is both diverse and accurate.

In summary, the overall flow of our framework is presented in Algorithm 1.

# 4. Experiments

In this section, we demonstrate the capability of TO-KENSWIFT in accelerating ultra-long sequences generation.

## 4.1. Setup

We conduct experiments on a variety of models, including `YaRN-LLaMA2-7b-128k` (Peng et al., 2024), `LLaMA3.1-8b` (AI@Meta, 2024) and `Qwen2.5-(1.5b,7b,14b)` (Qwen et al., 2025). For all models, we use the **Base** version, as the output length of Instruct version is limited (Bai et al., 2024). The inference experiments are performed on the test set of PG-19 (Rae et al., 2020).

**Training and Inference Details** We train linear layers in Section 3.2 using the first 8K tokens of training data, for datasets longer than 8K tokens, from PG-19 (Rae et al., 2020). The number of extra decoding heads is set to 3 across all models.

Inference is performed on a single NVIDIA A100-SXM4-80GB. When generating 100K tokens, the models are prefilled with 2K, 4K or 8K tokens as prompt from a random sample of the PG-19 test set (See Appendix F.1 for ablation on prefill length). The maximum budget of the partial cache is determined by the length of the prompt. For further training and inference details, please refer to Appendix B.

**Evaluation Metrics** We evaluate the overall *acceptance rate* and *speedup* for all methods. Unlike Leviathan et al. (2023)[3], our *acceptance rate* $\alpha$ is defined as:

$$\alpha = \frac{\sum_{i=1}^{T} a_i}{(\gamma + 1) \times T},\qquad(4)$$

where $a_i$ represents the number of tokens accepted at the $i$-th time step, $\gamma+1$ denotes the number of draft tokens generated at each time step, and $T$ represents the total number of time steps. The *speedup* is denotes as $\times$, which is the ratio of AR latency to TOKENSWIFT latency, given by:

$$\times = \frac{\text{latency}_{AR}}{\text{latency}_{\text{TOKENSWIFT}}},\qquad(5)$$

where latency refers to the average time required to generate a single token.

---

[3]The two can be converted into each other through computation.

We use *Distinct-n* (Li et al., 2016) to measure the diversity of generated content, *i.e.*, repetition. A higher value indicates greater diversity and lower repetition (Table 6).

**Baselines** We compare TOKENSWIFT with two baselines: **TriForce\***: The original TriForce (Sun et al., 2024a) employs a static KV update strategy, which cannot accelerate the generation of 100K tokens. The results in Table 3 correspond to our improved version of TriForce, which incorporates dynamic KV update [4]. **Medusa\***: To ensure losslessness, we adopt the Medusa (Cai et al., 2024) training recipe and incorporate the verification method of TOKENSWIFT. Both Medusa heads and tree structure are consistent with TOKENSWIFT.

The recent released MagicDec (Chen et al., 2024a) primarily focuses on acceleration for large throughput, and when the batch size is 1, `LLaMA3.1-8b` does not exhibit any acceleration for short text generation, let alone for ultra-long sequences. Therefore, it is excluded from our baseline.

## 4.2. Main Results

The experimental results are presented in Table 3 and Table 4. We evaluate TOKENSWIFT at different generation lengths of 20K, 40K, 60K, 80K and 100K, reporting *speedup* $\times$ and *acceptance rate* $\alpha$ by taking the average and standard deviation of 5 experiments to avoid randomness. Notably, the results for TOKENSWIFT and Medusa\* show a balanced trade-off between speed and quality, in contrast to TriForce\*, which suffers from low quality due to the absence of any repetition penalty.

**TOKENSWIFT significantly outperforms all baselines across generation lengths.** As shown in Table 3, across all lengths, TOKENSWIFT demonstrates superior acceleration performance compared to all baselines on models with different architectures (MHA, GQA). Moreover, TOKENSWIFT demonstrates remarkable robustness, showing virtually no impact when tested with varying prefix lengths.

**Longer generations amplify the speedup benefits.** As the generation length increases, the speed improvement of TOKENSWIFT becomes increasingly evident. Two key factors drive this trend: **Firstly**, AR experiences longer KV cache loading times as the number of tokens grows, whereas TOKENSWIFT mitigates this issue by utilizing dynamic KV pruning. **Secondly**, the acceptance rate improves as the number of tokens increases, primarily due to the higher $n$-grams acceptance rate. As the $n$-grams pool composed of generated tokens grows larger, the candidate $n$-grams become more diverse and accurate (Figure 3).

**Larger models yield greater speedup benefits.** The impact of frequent model reloading varies with model scale,

---

[4]To compare with `LLaMA3.1-8b`, we pretrained a draft model based on `LLaMA3.1-8b`. See Appendix C for details.

*Table 3.* Experimental results for LLaMA2 and LLaMA3.1 under varying prefix lengths, generating sequences from 20K to 100K tokens. $\alpha$ denotes the *acceptance rate* of all draft tokens (Equation (4)), while $\times$ represents the *speedup* ratio relative to AR (Equation (5)). TriForce* refers to our improved version, and Medusa* indicates the model we retrained (§4.1).

| Method | Gen. Len. | Prefill Len. 2048 | | Prefill Len. 4096 | | Prefill Len. 8192 | | Prefill Len. 2048 | | Prefill Len. 4096 | | Prefill Len. 8192 | |
|---|---|---|---|---|---|---|---|---|---|---|---|---|---|
| | | YaRN-LLaMA2-7b-128k (MHA) | | | | | | LLaMA3.1-8b (GQA) | | | | | |
| | | $\alpha$ | $\times(>1)$ | $\alpha$ | $\times(>1)$ | $\alpha$ | $\times(>1)$ | $\alpha$ | $\times(>1)$ | $\alpha$ | $\times(>1)$ | $\alpha$ | $\times(>1)$ |
| Medusa* | | 0.43 | 0.96 | 0.39 | 0.85 | 0.40 | 0.83 | 0.35 | 1.20 | 0.39 | 1.29 | 0.34 | 1.21 |
| TriForce* | **20K** | 0.80 | 1.50 | 0.89 | 1.51 | 0.92 | 1.36 | 0.89 | 1.13 | 0.89 | 1.08 | 0.99 | 1.16 |
| TOKENSWIFT | | $0.73_{\pm0.09}$ | $\mathbf{2.11}_{\pm0.14}$ | $0.68_{\pm0.09}$ | $\mathbf{2.02}_{\pm0.20}$ | $0.64_{\pm0.08}$ | $\mathbf{1.91}_{\pm0.12}$ | $0.64_{\pm0.08}$ | $\mathbf{1.87}_{\pm0.17}$ | $0.65_{\pm0.07}$ | $\mathbf{1.93}_{\pm0.18}$ | $0.72_{\pm0.09}$ | $\mathbf{1.99}_{\pm0.20}$ |
| Medusa* | | 0.52 | 1.08 | 0.42 | 0.86 | 0.43 | 0.88 | 0.35 | 1.26 | 0.40 | 1.39 | 0.34 | 1.26 |
| TriForce* | **40K** | 0.84 | 1.64 | 0.93 | 1.67 | 0.96 | 1.49 | 0.93 | 1.18 | 0.94 | 0.99 | 0.99 | 1.18 |
| TOKENSWIFT | | $0.82_{\pm0.06}$ | $\mathbf{2.60}_{\pm0.05}$ | $0.79_{\pm0.06}$ | $\mathbf{2.56}_{\pm0.09}$ | $0.79_{\pm0.05}$ | $\mathbf{2.50}_{\pm0.07}$ | $0.72_{\pm0.07}$ | $\mathbf{2.39}_{\pm0.16}$ | $0.73_{\pm0.08}$ | $\mathbf{2.47}_{\pm0.22}$ | $0.81_{\pm0.10}$ | $\mathbf{2.54}_{\pm0.22}$ |
| Medusa* | | 0.59 | 1.18 | 0.47 | 0.95 | 0.45 | 0.91 | 0.35 | 1.29 | 0.40 | 1.42 | 0.34 | 1.29 |
| TriForce* | **60K** | 0.85 | 1.76 | 0.95 | 1.83 | 0.97 | 1.62 | 0.94 | 1.21 | 0.95 | 0.96 | 1.00 | 1.19 |
| TOKENSWIFT | | $0.87_{\pm0.04}$ | $\mathbf{2.92}_{\pm0.04}$ | $0.85_{\pm0.04}$ | $\mathbf{2.89}_{\pm0.06}$ | $0.85_{\pm0.04}$ | $\mathbf{2.84}_{\pm0.05}$ | $0.75_{\pm0.06}$ | $\mathbf{2.73}_{\pm0.13}$ | $0.79_{\pm0.06}$ | $\mathbf{2.88}_{\pm0.17}$ | $0.85_{\pm0.08}$ | $\mathbf{2.93}_{\pm0.17}$ |
| Medusa* | | 0.61 | 1.17 | 0.51 | 0.99 | 0.47 | 0.93 | 0.35 | 1.30 | 0.40 | 1.43 | 0.34 | 1.29 |
| TriForce* | **80K** | 0.84 | 1.86 | 0.95 | 1.98 | 0.97 | 1.74 | 0.95 | 1.23 | 0.95 | 0.94 | 1.00 | 1.21 |
| TOKENSWIFT | | $0.89_{\pm0.03}$ | $\mathbf{3.13}_{\pm0.04}$ | $0.88_{\pm0.04}$ | $\mathbf{3.10}_{\pm0.06}$ | $0.88_{\pm0.03}$ | $\mathbf{3.05}_{\pm0.03}$ | $0.77_{\pm0.04}$ | $\mathbf{2.96}_{\pm0.07}$ | $0.82_{\pm0.06}$ | $\mathbf{3.13}_{\pm0.16}$ | $0.88_{\pm0.07}$ | $\mathbf{3.19}_{\pm0.13}$ |
| Medusa* | | 0.62 | 1.15 | 0.52 | 0.99 | 0.47 | 0.91 | 0.35 | 1.31 | 0.41 | 1.45 | 0.34 | 1.29 |
| TriForce* | **100K** | 0.82 | 1.94 | 0.96 | 2.14 | 0.97 | 1.86 | 0.95 | 1.25 | 0.96 | 0.92 | 0.99 | 1.22 |
| TOKENSWIFT | | $0.90_{\pm0.02}$ | $\mathbf{3.25}_{\pm0.05}$ | $0.90_{\pm0.03}$ | $\mathbf{3.23}_{\pm0.06}$ | $0.90_{\pm0.02}$ | $\mathbf{3.20}_{\pm0.02}$ | $0.79_{\pm0.03}$ | $\mathbf{3.13}_{\pm0.07}$ | $0.84_{\pm0.05}$ | $\mathbf{3.27}_{\pm0.19}$ | $0.90_{\pm0.06}$ | $\mathbf{3.38}_{\pm0.10}$ |

*Table 4.* Experimental results of TOKENSWIFT for Qwen2.5 across different scales under prefix length 4096, generating sequences from 20K to 100K tokens. $T_{AR}$ and $T_{\text{TOKENSWIFT}}$ denote the actual time required (in minutes) for AR and TOKENSWIFT, respectively. $\Delta_T$ represents the number of minutes saved by TOKENSWIFT compared to AR.

| Gen. Len. | Qwen2.5-1.5B | | | | | Qwen2.5-7B | | | | | Qwen2.5-14B | | | | |
|---|---|---|---|---|---|---|---|---|---|---|---|---|---|---|---|
| | $\alpha$ | $\times(>1)$ | $T_{AR}$ | $T_{\text{TOKENSWIFT}}$ | $\Delta_T$ | $\alpha$ | $\times(>1)$ | $T_{AR}$ | $T_{\text{TOKENSWIFT}}$ | $\Delta_T$ | $\alpha$ | $\times(>1)$ | $T_{AR}$ | $T_{\text{TOKENSWIFT}}$ | $\Delta_T$ |
| 20K | $0.69_{\pm0.11}$ | $1.69_{\pm0.17}$ | 12.00 | 7.20 | -4.80 | $0.64_{\pm0.07}$ | $2.00_{\pm0.16}$ | 15.60 | 7.80 | -7.80 | $0.67_{\pm0.06}$ | $2.12_{\pm0.13}$ | 29.40 | 13.80 | -15.60 |
| 40K | $0.80_{\pm0.06}$ | $2.31_{\pm0.09}$ | 36.00 | 15.60 | -20.40 | $0.77_{\pm0.05}$ | $2.64_{\pm0.10}$ | 47.40 | 18.00 | -29.40 | $0.78_{\pm0.03}$ | $2.68_{\pm0.10}$ | 89.40 | 33.60 | -55.80 |
| 60K | $0.85_{\pm0.04}$ | $2.69_{\pm0.07}$ | 73.80 | 27.60 | -46.20 | $0.78_{\pm0.08}$ | $2.86_{\pm0.25}$ | 95.40 | 33.60 | -61.80 | $0.82_{\pm0.02}$ | $3.01_{\pm0.13}$ | 184.20 | 61.20 | -123.00 |
| 80K | $0.87_{\pm0.03}$ | $2.95_{\pm0.06}$ | 124.20 | 42.00 | -82.20 | $0.80_{\pm0.09}$ | $3.07_{\pm0.30}$ | 161.40 | 52.80 | -108.60 | $0.83_{\pm0.02}$ | $3.20_{\pm0.13}$ | 312.60 | 97.80 | -214.80 |
| 100K | $0.89_{\pm0.07}$ | $3.13_{\pm0.07}$ | 187.80 | 60.00 | -127.80 | $0.82_{\pm0.09}$ | $3.23_{\pm0.28}$ | 244.20 | 75.60 | -168.60 | $0.84_{\pm0.02}$ | $3.34_{\pm0.10}$ | 474.60 | 142.20 | -332.40 |

as larger models require more time due to the increased parameters. As shown in Table 4, TOKENSWIFT demonstrates robust performance across models of different scales, with the acceleration advantage becoming more pronounced for larger models. In particular, when generating 100K tokens, TOKENSWIFT saves up to **5.54** hours for 14B model.

### 4.3. Ablation Studies

We conduct comprehensive ablation studies on TOKENSWIFT using LLaMA3.1-8b. For all experiments, the prefix length is 4096.

#### 4.3.1. TOKEN REUTILIZATION

We define the *n-gram acceptance rate* $\beta$ similarly to Equation (4). Let $a_i'$ denote the length of accepted *n*-gram candidate at iteration $i$. Then $\beta$ is given by:

$$\beta = \frac{\sum_{i=1}^{T} b_i}{(\gamma + 1) \times T}, \quad \text{where,} \ b_i = \begin{cases} a_i', & a_i' = a_i \\ 0, & a_i' < a_i \end{cases}. \quad (6)$$

From Figure 3, we observe that removing token reutilization ($k = 0$) leads to a significant decrease in both *acceptance rate* $\alpha$ and *speedup* $\times$. Furthermore, as the generation length increases, the *acceptance rate* $\alpha$ for $k = 0$ slightly drops.

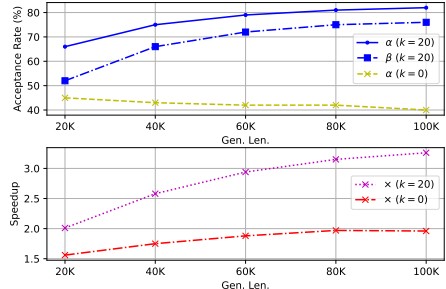

*Figure 3.* Upper: The *acceptance rate* $\alpha$ for $k = 20$ and $k = 0$, along with the *n-gram acceptance rate* $\beta$ for $k = 20$, plotted against varying generation lengths. Lower: The *speedup* $\times$ achieved at different generation lengths.

This trend stems from the fact that, in ultra-long sequences, the KV cache cannot be compressed indefinitely. In contrast, TOKENSWIFT ($k = 20$) shows an increasing *acceptance rate* as the sequence length grows, demonstrating the effectiveness of token reutilization in **reducing the frequency of model reloading**.

#### 4.3.2. DYNAMIC KV UPDATES

To evaluate the effectiveness of TOKENSWIFT's dynamic KV update policy, we experiment with three different strate-

gies of managing KV cache during drafting:

• Full Cache: Retaining full KV cache throughout drafting.

• Partial Cache: Updating partial KV cache only once during the prefill phase.

• Dynamic Partial Cache: Dynamically updating KV cache as described in §3.3

For a fair comparison, token reutilization is disabled (*i.e.* $k = 0$). As shown in Table 5, Partial Cache leads to a low acceptance rate, resulting in reduced speedup. While Full Cache achieves a higher acceptance rate, its computational overhead negates the speedup gains. In contrast, Dynamic Partial Cache adopted by TOKENSWIFT strikes a balanced trade-off, achieving both high acceptance rate and significant speedup. As a result, Dynamic Partial Cache can **effectively manage partial KV under ultra-long sequence generation.**

*Table 5.* The ablation experiment results on *KV management*.

| Gen. Len. | | Full Cache | Partial Cache | Dynamic Partial Cache |
|---|---|---|---|---|
| **20K** | $\alpha$ | 0.42 | 0.19 | 0.45 |
| | $\times(>1)$ | 1.36 | 0.94 | **1.56** |
| **40K** | $\alpha$ | 0.42 | 0.16 | 0.43 |
| | $\times(>1)$ | 1.42 | 1.03 | **1.75** |
| **60K** | $\alpha$ | 0.42 | 0.18 | 0.42 |
| | $\times(>1)$ | 1.45 | 1.19 | **1.88** |
| **80K** | $\alpha$ | 0.42 | 0.19 | 0.42 |
| | $\times(>1)$ | 1.46 | 1.31 | **1.97** |
| **100K** | $\alpha$ | 0.42 | 0.21 | 0.40 |
| | $\times(>1)$ | 1.47 | 1.44 | **1.96** |

*Table 6.* The ablation experiment results on *contextual penalty* using different sampling methods. Light cell represents the settings adopted by TOKENSWIFT. We take $\theta = 1.2, W = 1024$.

| | Distinct-1 | Distinct-2 | Distinct-3 | Distinct-4 | AVG. | $\times$ |
|---|---|---|---|---|---|---|
| **top-p** | 0.15 | 0.25 | 0.29 | 0.31 | **0.25** | 3.42 |
| *w/o. penalty* | 0.09 | 0.15 | 0.18 | 0.20 | 0.16 | 3.53 |
| **$\eta$-sampling** | 0.25 | 0.43 | 0.49 | 0.53 | **0.43** | 3.42 |
| *w/o. penalty* | 0.06 | 0.10 | 0.12 | 0.13 | 0.11 | 3.57 |
| **min-p** | 0.41 | 0.71 | 0.81 | 0.82 | **0.69** | 3.27 |
| *w/o. penalty* | 0.07 | 0.11 | 0.14 | 0.15 | 0.12 | 3.58 |

### 4.3.3. CONTEXTUAL PENALTY

As an orthogonal method to min-$p$, top-$p$, and $\eta$-sampling for mitigating the repetition, *contextual penalty* demonstrates effectiveness across different sampling methods.

As shown in Table 6, without *contextual penalty*, the diversity of generated sequences is significantly lower for all sampling methods. The most striking improvement emerges in min-$p$ sampling (See Appendix D for more sampling details), where the average Distinct-$n$ score surges from 0.12 to 0.69 with only an 8% compromise in speedup. These results clearly highlight the impact of contextual penalty in **mitigating repetitive token generation.** It can seamlessly integrate with existing sampling methods to enhance the

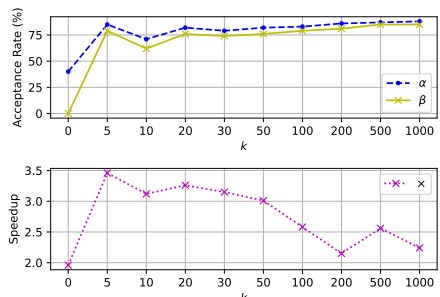

*Figure 4.* Upper: The *acceptance rate* $\alpha$ and *n-gram acceptance rate* $\beta$ versus varying $k$. Lower: The *speedup* $\times$ versus varying $k$.

*Table 7.* Acceptance rate $\alpha$ ($k = 0$) and *speedup* $\times$ across different tree configurations. Each configuration is represented by a 4-digit array: they represent the number of candidates for different decoding heads in §3.2.

| Gen. Len. | | [3,3,3,3] | [1,9,9,9] | [1,3,3,3] (Ours) |
|---|---|---|---|---|
| **20K** | $\alpha$ | 0.44 | 0.50 | 0.45 |
| | $\times(>1)$ | 1.34 | 0.53 | **1.56** |
| **40K** | $\alpha$ | 0.43 | 0.52 | 0.43 |
| | $\times(>1)$ | 1.58 | 0.67 | **1.75** |
| **60K** | $\alpha$ | 0.43 | 0.53 | 0.42 |
| | $\times(>1)$ | 1.75 | 0.78 | **1.88** |
| **80K** | $\alpha$ | 0.43 | 0.55 | 0.42 |
| | $\times(>1)$ | 1.85 | 0.88 | **1.97** |
| **100K** | $\alpha$ | 0.42 | 0.57 | 0.40 |
| | $\times(>1)$ | 1.91 | 0.96 | **1.96** |

*Table 8.* Distinct-$n$ score across different penalty value $\theta$. 1.0 indicate that no penalty is applied. We take $W = 1024$ (See Appendix F.2 for ablation on $W$).

| $\theta$ | Distinct-1 | Distinct-2 | Distinct-3 | Distinct-4 | AVG. |
|---|---|---|---|---|---|
| **1.0** | 0.07 | 0.11 | 0.14 | 0.15 | 0.12 |
| **1.1** | 0.08 | 0.13 | 0.15 | 0.16 | 0.13 |
| **1.2** | 0.41 | 0.71 | 0.81 | 0.82 | 0.69 |
| **1.3** | 0.57 | 0.86 | 0.93 | 0.95 | 0.83 |
| **1.4** | 0.52 | 0.73 | 0.76 | 0.77 | 0.70 |
| **1.5** | 0.74 | 0.96 | 0.98 | 0.99 | 0.92 |

quality of ultra-long sequence generation.

In addition, we can find that the higher the diversity, the lower the *speedup*. Therefore, if TriForce is combined with *context penalty*, the *speedup* in Table 3 will drop further.

### 4.4. Discussions

In this section, we explore the effects of different hyperparameters on TOKENSWIFT.

#### 4.4.1. TREE CONFIGURATION

Due to the time-consuming nature of finding the optimal tree in Medusa (Cai et al., 2024) and its limited impact on

accelerating ultra-long sequences generation, we employ a simple 3-ary tree in tree attention. See Appendix B for the tree structure.

As shown in Table 7, `[1,9,9,9]` has the highest acceptance rate but the lowest speedup. This is because more candidates increase the acceptance rate, but also increase the verification burden. Similarly, by comparing `[1,3,3,3]` and `[3,3,3,3]`, we can find that the first head (*i.e.*, the original head of target model) achieves relatively high prediction accuracy when using KV compression, so choosing the top-1 token as candidate is sufficient. To balance the trade-off of acceptance rate and verification efficiency, we adopt `[1,3,3,3]` as the configuration of TOKENSWIFT.

### 4.4.2. N-GRAM CANDIDATES

As illustrated in Figure 4, increasing $k$ enhances the *n-gram acceptance rate* $\beta$ due to a larger pool of $n$-gram candidates. However, an excessive number of candidates can strain the verification process, leading to reduced *speedup* $\times$.

Interestingly, a lower $k$ does not always result in a lower $\beta$. For instance, $k = 5$ achieves a higher $\beta$ than $k = 20$, resulting in both a higher *acceptance rate* $\alpha$ and greater *speedup* $\times$. However, at $k = 5$, the lack of diversity among the candidates leads to increased repetition, which in turn degrades the quality of generation.

### 4.4.3. PENALTY VALUE $\theta$

As a key component of TOKENSWIFT, *contextual penalty* significantly reduces repetition in generated text. We examine the effect of two parameters present in contextual penalty, *i.e.* penalty value $\theta$ and penalty window $W$.

Table 8 presents the impact of introducing contextual penalty on diversity. Without any penalty ($\theta = 1.0$), the generated sequences exhibit severe repetition, with an average Distinct-$n$ score of only **0.12**. As the value of $\theta$ increases gradually to 1.2, the diversity improves significantly, highlighting the effectiveness of contextual penalty in enhancing the diversity of ultra-long sequence generation.

### 4.5. Temperature

Table 9 presents the results of an ablation experiment investigating the effect of varying temperature settings on the generation length, acceptance rate, and speedup during text generation. The experiment uses top-$p$ sampling with a fixed $p$ of 0.9 and evaluates generation lengths ranging from 20K to 100K tokens, with temperature values spanning from 0.4 to 1.2.

From the results, it is evident that as temperature increases, acceptance rate generally decreases across all generation lengths. Specifically, acceptance rate drops from 0.79 at a temperature of 0.4 to 0.52 at a temperature of 1.2 for 20K-length generation, and a similar trend is observed for longer

sequences. This suggests that higher temperatures result in more diverse but less accurate output. On the other hand, speedup tends to remain relatively stable or slightly decrease with higher temperatures. The highest speedups, reaching around 3.4, are observed across all generation lengths with temperatures around 0.6 and 1.0, indicating that moderate temperature settings offer the best balance between speed and quality.

*Table 9.* Ablation results on varying temperatures. Using top-$p$ sampling, with $p$ set to 0.9.

| Gen. Len. | | 0.4 | 0.6 | 0.8 | 1.0 | 1.2 |
|---|---|---|---|---|---|---|
| 20K | $\alpha$ | 0.79 | 0.84 | 0.56 | 0.68 | 0.52 |
| | $\times (> 1)$ | 2.25 | 2.34 | 1.80 | 2.10 | 1.72 |
| 40K | $\alpha$ | 0.85 | 0.88 | 0.73 | 0.81 | 0.69 |
| | $\times (> 1)$ | 2.76 | 2.80 | 2.60 | 2.80 | 2.52 |
| 60K | $\alpha$ | 0.87 | 0.89 | 0.80 | 0.84 | 0.77 |
| | $\times (> 1)$ | 3.07 | 3.10 | 3.05 | 3.07 | 2.96 |
| 80K | $\alpha$ | 0.88 | 0.90 | 0.83 | 0.86 | 0.81 |
| | $\times (> 1)$ | 3.26 | 3.29 | 3.29 | 3.28 | 3.22 |
| 100K | $\alpha$ | 0.89 | 0.90 | 0.85 | 0.87 | 0.83 |
| | $\times (> 1)$ | 3.39 | 3.41 | 3.45 | 3.42 | 3.42 |

### 4.5.1. CASE STUDY

Figure 5 presents a case study on the impact of the *contextual penalty*. Without the Contextual Penalty, repetitions appear at about 5K tokens, compared to 60K with the penalty applied. Additionally, generation without the penalty exhibits word-for-word repetition, whereas generation with the penalty primarily demonstrates semantic-level repetition, highlighting its effectiveness in mitigating redundancy.

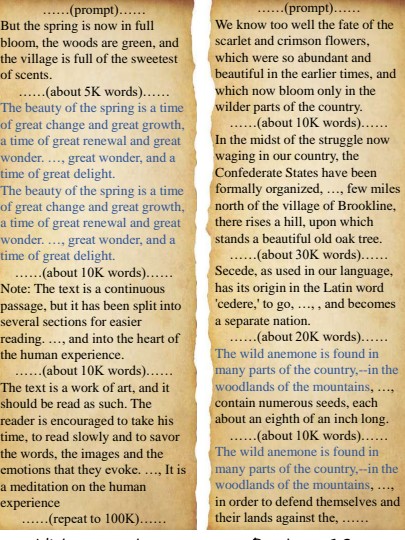

*Figure 5.* Case Study on `LLaMA3.1-8b`. Left: fragments of generated text without Contextual Penalty. Right: fragments of generated text with Contextual Penalty. The blue text is repetition part. See Appendix G for more cases.

# 5. Related Works

## 5.1. Speculative Decoding

Recent advancements in speculative decoding have significantly accelerated large language model (LLM) inference through diverse methodologies. Speculative decoding (Leviathan et al., 2023; Chen et al., 2023) traditionally leverages smaller draft models to propose candidate tokens for verification by the target model. Early works like SpecTr (Sun et al., 2023) introduced optimal transport for multi-candidate selection, while SpecInfer (Miao et al., 2024) and Medusa (Cai et al., 2024) pioneered tree-based structures with tree-aware attention and multi-head decoding to enable parallel verification of multiple candidates. Subsequent innovations, such as Sequoia (Chen et al., 2024b) and EAGLE-2 (Li et al., 2024d), optimized tree construction using dynamic programming and reordering strategies, while Hydra (Ankner et al., 2024) and ReDrafter (Cheng et al., 2024) enhanced tree dependencies through sequential or recurrent heads. Hardware-aware optimizations, exemplified by SpecExec (Svirschevski et al., 2024) and Triforce (Sun et al., 2024a), further improved efficiency by leveraging hierarchical KV caching and quantized inference.

Self-speculative approaches eliminate the need for external draft models by exploiting internal model dynamics. Draft&Verify (Zhang et al., 2024) and LayerSkip (Elhoushi et al., 2024) utilized early-exit mechanisms and Bayesian optimization to skip layers adaptively, whereas Kangaroo (Liu et al., 2024b) integrated dual early exits with lightweight adapters. Sun et al. (2024b) and SpecDec++ (Huang et al., 2024) introduced theoretical frameworks for block-level token acceptance and adaptive candidate lengths. Parallel decoding paradigms, such as PASS (Monea et al., 2023) and MTJD (Qin et al., 2024), employed look-ahead embeddings or joint probability modeling to generate multiple candidates in a single pass, while CLLMs (Kou et al., 2024) and Lookahead (Fu et al., 2024) reimagined autoregressive consistency through Jacobi decoding and n-gram candidate pools.

Retrieval-augmented methods like REST (He et al., 2024), and NEST (Li et al., 2024b) integrated vector or phrase retrieval to draft context-aware tokens, often combining copy mechanisms with confidence-based attribution. Training-centric strategies, including TR-Jacobi (Wang et al., 2024a), enhanced parallel decoding capability via noisy training or self-distilled multi-head architectures. System-level optimizations such as PipeInfer (Butler et al., 2024) and Narasimhan et al. (2024) addressed scalability through asynchronous pipelines and latency-aware scheduling, while Goodput (Liu et al., 2024c) focused on dynamic resource allocation and nested model deployment.

Approaches such as Triforce (Sun et al., 2024a) and MagicDec (Chen et al., 2024a) incorporate KV cache compression during the drafting phase. However, their applicability is limited to scenarios characterized by long prefixes and short outputs, making them unsuitable for ultra-long sequence generation tasks. In such tasks, which are the focus of our work, the need for efficient inference spans both extended input contexts and lengthy outputs, presenting challenges that existing methods fail to address.

## 5.2. Long Sequence Generation

Recent advances in long sequence generation have focused on addressing the challenges of coherence, efficiency, and scalability in producing extended outputs. A pivotal contribution is the LongWriter (Bai et al., 2024) framework, which introduces a task decomposition strategy to generate texts exceeding 20,000 words. Complementing this, Temp-Lora (Wang et al., 2024b) proposes inference-time training with temporary Lora modules to dynamically adapt model parameters during generation, offering a scalable alternative to traditional KV caching. Similarly, PLANET (Hu et al., 2022) leverages dynamic content planning with sentence-level bag-of-words objectives to improve logical coherence in opinion articles and argumentative essays, demonstrating the effectiveness of structured planning in autoregressive transformers.

In addition, lightweight decoding-side sampling strategies have emerged for repetition mitigation. The foundational work on Nucleus Sampling (Holtzman et al., 2020) first demonstrated that dynamically truncating low-probability token sets could reduce repetitive outputs while maintaining tractable decoding latency. Building on this, Hewitt et al. (2022) introduced $\eta$-sampling explicitly linking candidate set reduction to repetition mitigation by entropy-guided token pruning. Recent variants like Min-p (Nguyen et al., 2024) optimize truncation rules in real-time—scaling thresholds to the maximum token probability. And Mirostat Sampling (Basu et al., 2021) further integrate lightweight Bayesian controllers to adjust $\eta$ parameters on-the-fly. Our work systematically analyzing how parameterized sampling (*e.g.*, Top-p Min-p, $\eta$-sampling) balances computational overhead and repetition suppression in ultra-long sequence generation pipelines.

# 6. Conclusion

In this study, we introduce TOKENSWIFT, a novel framework designed to achieve lossless acceleration in generating ultra-long sequences with LLMs. By analyzing and addressing three challenges, TOKENSWIFT significantly enhances the efficiency of the generation process. Our experimental results demonstrate that TOKENSWIFT achieves over $3\times$ acceleration across various model scales and architectures. Furthermore, TOKENSWIFT effectively mitigates issues related to repetitive content, ensuring the quality and coherence of the generated sequences.

## Impact Statement

This paper presents work aimed at advancing the field of Machine Learning, specifically in the context of improving efficiency and scalability in generating ultra-long sequences. There are many potential societal consequences of our work, none which we feel must be specifically highlighted here. None of the ethical concerns we foresee require specific actions or warnings in the context of this work.

## Acknowledgement

The authors thank the reviewers for their insightful suggestions to improve the manuscript. This work presented herein is supported by the National Natural Science Foundation of China (62376031).

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

# A. Lossless Nature of Speculative Decoding

The speculative decoding (Leviathan et al., 2023; Chen et al., 2023) can easily be justified to be lossless and identical to sample from $q_{target}$ alone, *i.e.*, $p_{SD} = q_{target}$. Note that, given prefix $X_{1:j}$, the next token sampled from:

$$x_{j+1} \sim \begin{cases} p_{draft}(x|X_{1:j}), & \text{if } \mathcal{U}(0,1) > \alpha, \\ norm(\max(0, q_{target}(x|X_{1:j}) - p_{draft}(\hat{x}|X_{1:j}))), & \text{otherwise}, \end{cases}$$

where $\alpha$ is the acceptance rate given by

$$\alpha(x) = \min\left(1.0, \frac{q_{target}(x)}{p_{draft}(x)}\right).$$

If the draft token is accepted, we have

$$p_{SD}(x|X_{1:j}; accepted) = p_{draft}(x|X_{1:j})\alpha(x|X_{1:j}) = \min(p_{draft}, q_{target}).$$

If the token is rejected, we have

$$p_{SD}(x|X_{1:j}; rejected) = (1 - \alpha(x|X_{1:j}))norm(\max(0, q_{target}(x|X_{1:j}) - p_{draft}(\hat{x}|X_{1:j})))$$
$$= (1 - \alpha)\frac{q_{target} - \min(p_{draft}, q_{target})}{1 - \alpha}$$
$$= q_{target} - \min(p_{draft}, q_{target})$$

Therefore, the overall probability is given by

$$p_{SD}(x|X_{1:j}) = p_{SD}(x|X_{1:j}; accepted) + p_{SD}(x|X_{1:j}; rejected) = q_{target}((x|X_{1:j})$$

Proved.

# B. Additional Training and Inference Details.

## B.1. Training Details

During training, only three linear layers are fine-tuned, while the parameters of the LLM remained fixed. The model was trained on an NVIDIA A100-SXM4-80GB GPU. The specific training parameters are outlined in Table 10.

*Table 10.* Additional training details. Note that these hyperparameters do not require extensive tuning.

|  | LLaMA3.1-8b | YaRN-LLaMA2-7b-128k | Qwen2.5-1.5b | Qwen2.5-7b | Qwen2.5-14b |
|---|---|---|---|---|---|
| optimizer | | | AdamW | | |
| betas | | | (0.9, 0.999) | | |
| weight decay | | | 0.1 | | |
| warmup steps | | | 50 | | |
| learning rate scheduler | | | cosine | | |
| num. GPUs | | | 4 | | |
| gradient accumulation steps | | | 10 | | |
| batch size per GPU | | | 3 | | 1 |
| num. steps | | | 200 | | 600 |
| learning rate | | | 5e-3 | | 1e-3 |
| training parameters | 50.3M | 50.3M | 7.1M | 38.5M | 78.6M |
| MAX kv cache size | 13.4G | 53.7G | 2.9G | 5.9G | 20.1G |

## B.2. Inference Details

For inference, we used 4-grams to maintain consistency with multi-token generation. The specific inference parameters are presented in Table 11.

For the tree attention mechanism, we selected a simple ternary full tree configuration, as depicted in Appendix B.2.

*Table 11.* $k$ stands for the maximum number of retrieved n-grams in token reutilization

|  | $k$ | temp. | top-$p$ | min-$p$ | penalty | penalty len. |
|---|---|---|---|---|---|---|
| LLaMA3.1-8b |  |  | - | 0.1 | 1.2 |  |
| YaRN-LLaMA2-7b-128k |  |  | 0.9 | - | 1.15 |  |
| Qwen2.5-1.5b | 20 | 1.0 | 0.9 | - | 1.15 | 1024 |
| Qwen2.5-7b |  |  | - | 0.05 | 1.15 |  |
| Qwen2.5-14b |  |  | - | 0.05 | 1.13 |  |

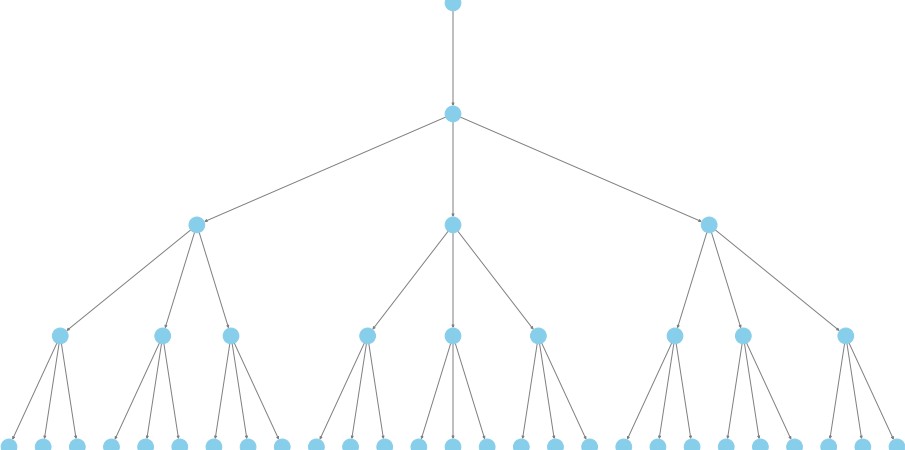

## C. Pre-training Details of the Llama3.1 Draft Model

To serve as the draft model for `LLaMA3.1-8b` in TriForce, we pretrain a tiny version of 250M parameters with the same tokenizer from `LLaMA3.1-8b`. The model configuration is listed in Table 12. We train the model on Wikipedia (20231101.en) [5] and part of C4-en[6] for 1 epoch.

*Table 12.* Configuration of Llama 3.1 205M.

| | |
|---|---|
| hidden_size | 768 |
| hidden_act | silu |
| intermediate_size | 3072 |
| max_position_embeddings | 2048 |
| num_attention_heads | 12 |
| num_key_value_heads | 12 |
| rope_theta | 500000 |
| vocab_size | 128256 |

## D. Different Sampling Method

### D.1. Introduction of Different Sampling Algorithms

Given a probability distribution $P(x_t|x_1, x_2, \ldots, x_{t-1})$ over the vocabulary $\mathcal{V}$ at position $t$, top-$p$ sampling (Holtzman et al., 2020) first sorts the tokens in descending order of their probabilities. It then selects the smallest set of tokens whose cumulative probability exceeds a predefined threshold $p$, where $p \in (0, 1]$. Formally, let $\mathcal{V}_p \subset \mathcal{V}$ be the smallest set such that:

$$\sum_{v \in \mathcal{V}_p} P(x_t = v|x_1, x_2, \ldots, x_{t-1}) \geqslant p.$$

---

[5] https://huggingface.co/datasets/wikimedia/wikipedia
[6] https://huggingface.co/datasets/allenai/c4

The next token $\hat{x}_t$ is then randomly sampled from this reduced set $\mathcal{V}_p$ according to the renormalized probabilities:

$$\hat{x}_t \sim \frac{P(x_t = v|x_1, \ldots, x_{t-1})}{\sum_{v' \in \mathcal{V}_p} P(x_t = v'|x_1, \ldots, x_{t-1})} \text{ for } v \in \mathcal{V}_p.$$

Nguyen et al. (2024) introduced min-p sampling, which uses a relative probability threshold $p_{base} \in (0, 1]$ to scale the maximum token probability $p_{max}$ to determine the absolute probability threshold $p_{scaled}$. Sampling is then performed on tokens with probability greater than or equal to $p_{scaled}$.

Formally, given the maximum probability over the token distribution $p_{max} = \max_{v \in \mathcal{V}} P(x_t = v|x_1, x_2, \ldots, x_{t-1})$, the absolute probability threshold $p_{scaled}$ is calculated as:

$$p_{scaled} = p_{base} \times p_{max}.$$

The sampling pool $\mathcal{V}_{min}$ is then defined as the set of tokens whose probability is greater than or equal to $p_{scaled}$:

$$\mathcal{V}_{min} = \{v \in \mathcal{V} \mid P(v|x_1, x_2, \ldots, x_{t-1}) \geqslant p_{scaled}\}.$$

Finally, the next token $\hat{x}_t$ is randomly sampled from the set $\mathcal{V}_{min}$ according to the normalized probabilities:

$$\hat{x}_t \sim \frac{P(v|x_1, \ldots, x_{t-1})}{\sum_{v' \in \mathcal{V}_{min}} P(v'|x_1, \ldots, x_{t-1})} \text{ for } v \in \mathcal{V}_{min}.$$

The sampling pool of $\eta$-sampling (Hewitt et al., 2022) is defined as

$$\mathcal{V}_\eta = \{v \in \mathcal{V} \mid P(v|x_1, x_2, \ldots, x_{t-1}) \geqslant \eta\},$$
$$\eta = \min\left(\epsilon, \alpha \exp(-h_{\theta, x_{<i}})\right).$$

where $h_{\theta, x_{<i}}$ is the entropy of $P(\mathcal{V}|x_1, x_2, \ldots, x_{t-1})$, $\alpha$ and $\epsilon$ are hyperparameters.

### D.2. Impact of Different Sampling Algorithms

We also explored the impact of different sampling algorithms with disable token reutilization, including top-$p$ sampling (Holtzman et al., 2020), min-$p$ sampling (Nguyen et al., 2024), and $\eta$-sampling (Hewitt et al., 2022). As summarized in Table 13, TOKENSWIFT consistently demonstrates strong robustness across these methods. This versatility underscores its compatibility with a wide range of decoding strategies, making it suitable for diverse applications and use cases.

*Table 13.* Ablation results on various sampling methods with disable token reutilization.

| Gen. Len. | | top-$p$ $(p = 0.9)$ | min-$p$ $(p = 0.1)$ | $\eta$-sampling $(\epsilon = $2e-4$)$ |
|---|---|---|---|---|
| **20K** | $\alpha$ | 0.68 | 0.66 | 0.56 |
| | $\times (> 1)$ | 2.10 | 2.01 | 1.85 |
| **40K** | $\alpha$ | 0.81 | 0.75 | 0.71 |
| | $\times (> 1)$ | 2.80 | 2.58 | 2.59 |
| **60K** | $\alpha$ | 0.84 | 0.79 | 0.78 |
| | $\times (> 1)$ | 3.07 | 2.94 | 2.99 |
| **80K** | $\alpha$ | 0.86 | 0.81 | 0.81 |
| | $\times (> 1)$ | 3.28 | 3.15 | 3.24 |
| **100K** | $\alpha$ | 0.87 | 0.82 | 0.84 |
| | $\times (> 1)$ | 3.42 | 3.26 | 3.42 |

# E. Tree-Based Attention

Tree attention is a mechanism designed to process multiple candidate continuations during speculative decoding efficiently. Instead of selecting a single continuation as in traditional methods, tree attention leverages multiple candidates to increase the expected acceptance length in each decoding step, balancing computational demands and performance.

The mechanism uses a tree structure where each branch represents a unique candidate continuation. For example, if two heads generate top-2 and top-3 predictions, the Cartesian product of these predictions results in 6 candidates, forming a tree with 6 branches. Each token in the tree attends only to its predecessors, and an attention mask ensures that this constraint is upheld. Positional indices are also adjusted to align with the tree structure.

The tree structure is constructed by taking the Cartesian product of the predictions across all heads. If head $k$ has $s_k$ top predictions, then the tree structure consists of all possible combinations of predictions across the heads. Each combination forms a unique branch in the tree.

Let the total number of candidates (i.e., branches) in the tree be denoted as $C$, which is the product of the number of predictions for each head:

$$C = \prod_{k=1}^{K} s_k.$$

Each candidate is a distinct sequence of tokens formed by selecting one token from each set of predictions from the heads.

To ensure that tokens only attend to their predecessors (tokens generated earlier in the continuation), an attention mask is applied. The attention mask for the tree structure ensures that for each token at level $k$, it can attend only to tokens in levels $\{0, 1, \ldots, k-1\}$. This guarantees that each token's attention is directed solely towards its predecessors in the tree.

Formally, the attention mask $M_k$ for each token at level $k$ is defined as:

$$M_k(i, j) = \begin{cases} 1 & \text{if token } j \text{ is a predecessor of token } i, \\ 0 & \text{otherwise.} \end{cases}$$

where $M_k(i, j) = 1$ means that the token at position $j$ can attend to the token at position $i$, and $M_k(i, j) = 0$ means no attention is allowed from $j$ to $i$.

# F. More Ablation Experiments

## F.1. Ablation of Prefill Length

We disable token reutilization and conduct ablation study on the different prefix length, as shown in Table 14. The experiment explores the impact of varying prefix lengths on the generation of sequences of different lengths (from 20K to 100K). The results include two key metrics: acceptance rate ($\alpha$) and speedup factor ($\times$).

As the prefix length increases, the acceptance rate tends to stabilize, generally hovering around 0.35 to 0.39 across different sequence lengths, with a slight fluctuation depending on the specific prefix length. This suggests that while the acceptance rate does not dramatically change with longer sequences, it remains relatively consistent.

*Table 14.* Ablation results on different prefill length disable token reutilization.

| Prefill Len. | 20K | | 40K | | 60K | | 80K | | 100K | |
|---|---|---|---|---|---|---|---|---|---|---|
| | $\alpha$ | $\times$ | $\alpha$ | $\times$ | $\alpha$ | $\times$ | $\alpha$ | $\times$ | $\alpha$ | $\times$ |
| 2048 | 0.35 | 1.41 | 0.35 | 1.63 | 0.35 | 1.76 | 0.35 | 1.83 | 0.34 | 1.87 |
| 3072 | 0.31 | 1.23 | 0.31 | 1.42 | 0.31 | 1.55 | 0.31 | 1.64 | 0.30 | 1.69 |
| 4096 | 0.35 | 1.32 | 0.35 | 1.54 | 0.35 | 1.69 | 0.35 | 1.76 | 0.35 | 1.85 |
| 5120 | 0.32 | 1.29 | 0.31 | 1.46 | 0.31 | 1.57 | 0.31 | 1.65 | 0.31 | 1.70 |
| 6144 | 0.39 | 1.46 | 0.39 | 1.66 | 0.39 | 1.80 | 0.39 | 1.88 | 0.39 | 1.94 |
| 7168 | 0.36 | 1.42 | 0.37 | 1.62 | 0.36 | 1.74 | 0.36 | 1.82 | 0.36 | 1.88 |
| 8192 | 0.36 | 1.21 | 0.36 | 1.42 | 0.36 | 1.58 | 0.36 | 1.69 | 0.36 | 1.77 |

In terms of speedup, it shows that with longer prefix lengths, the model achieves progressively higher acceleration. For instance, a prefix length of 2048 achieves a speedup of 1.41 for 20K tokens, but with 8192, the speedup reaches up to 1.77

for 100K tokens. This indicates that increasing the prefix length contributes to better acceleration, especially for longer sequences, while maintaining a relatively stable acceptance rate. The findings demonstrate the tradeoff between prefix length and model efficiency, where larger prefix lengths tend to result in greater speed.

### F.2. Ablation of Penalty Window

We investigate the effect of penalty window size ($W$) on the performance of a model generating sequences of varying lengths (from 20K to 100K tokens). For each sequence length, we apply a penalty to generated tokens within a sliding window of size $W$, and evaluate the impact on two key metrics: acceptance rate ($\alpha$) and acceleration factor ($\times$). Additionally, we assess the diversity of the generated sequences using the *Distinct-n* metric, where higher values indicate greater diversity.

*Table 15.* Ablation results on penalty length ($W$).

| Penalty Len. ($W$) | 20K | | 40K | | 60K | | 80K | | 100K | |
|---|---|---|---|---|---|---|---|---|---|---|
| | $\alpha$ | $\times$ | $\alpha$ | $\times$ | $\alpha$ | $\times$ | $\alpha$ | $\times$ | $\alpha$ | $\times$ |
| 20 | 0.82 | 2.25 | 0.90 | 2.85 | 0.93 | 3.20 | 0.94 | 3.42 | 0.95 | 3.58 |
| 50 | 0.83 | 2.30 | 0.89 | 2.83 | 0.91 | 3.14 | 0.92 | 3.35 | 0.93 | 3.52 |
| 128 | 0.59 | 1.75 | 0.70 | 2.38 | 0.75 | 2.75 | 0.80 | 3.07 | 0.82 | 3.29 |
| 256 | 0.78 | 2.17 | 0.86 | 2.76 | 0.89 | 3.11 | 0.91 | 3.33 | 0.92 | 3.48 |
| 512 | 0.75 | 2.15 | 0.84 | 2.73 | 0.88 | 3.07 | 0.89 | 3.28 | 0.90 | 3.43 |
| 1024 | 0.66 | 2.01 | 0.75 | 2.58 | 0.79 | 2.94 | 0.81 | 3.15 | 0.82 | 3.26 |
| 2048 | 0.69 | 1.99 | 0.79 | 2.58 | 0.82 | 2.91 | 0.84 | 3.14 | 0.86 | 3.31 |

*Table 16.* Distinct-$n$ score with different penalty length $W$.

| Penalty Len. ($W$) | Distinct-1 | Distinct-2 | Distinct-3 | Distinct-4 | AVG. |
|---|---|---|---|---|---|
| 20 | 0.85 | 0.86 | 0.73 | 0.70 | 0.79 |
| 50 | 0.91 | 0.91 | 0.85 | 0.77 | 0.86 |
| 128 | 0.95 | 0.77 | 0.57 | 0.48 | 0.69 |
| 256 | 0.83 | 0.91 | 0.88 | 0.83 | 0.86 |
| 512 | 0.90 | 0.86 | 0.74 | 0.65 | 0.79 |
| 1024 | 0.79 | 0.86 | 0.77 | 0.71 | 0.78 |
| 2048 | 0.67 | 0.84 | 0.86 | 0.84 | 0.80 |

The results in Table 15 and Table 16 show a clear trade-off between the penalty window size and the model's performance. For smaller penalty window sizes, such as $W = 20$, the model achieves higher acceptance rates and better acceleration, but this comes at the cost of lower diversity in the generated sequences (as indicated by lower *Distinct-n* values). As the penalty window size increases (*e.g.*, $W = 256$ or $W = 2048$), the acceptance rate slightly decreases, but the model exhibits better diversity and still maintains a significant speedup relative to the AR baseline. These findings suggest that larger penalty windows can help reduce repetitiveness and improve the diversity of long sequence generation, but they may also slightly reduce the model's efficiency and acceptance rate.

Table 15 also reveals that for each penalty window size, increasing the sequence length (from 20K to 100K tokens) generally results in higher acceleration and better diversity, with some fluctuations in acceptance rates.

## G. More Cases

## H. Training Loss Curve

……(prompt)……
of the literature which
has been devoted to him.

The Revolutionary Epoch in
Europe.

If we examine the last half of the
eighteenth century in Europe, we
find that it was marked by the
most extraordinary commotion
……(about 5K words)……
In the eighteenth century, the
Revolution was the result of the
revolt of the mob, and the Reign
of Terror was the result of the
revolt of the mob.
The American Revolution was the
result of the revolt of the
Revolutionary mob, and the
Reign of Terror was the result of
the revolt of the Revolutionary
mob.
……(about 55K words)……
In the eighteenth century, the
Revolution was the result of the
revolt of the mob, and the Reign
of Terror was the result of the
revolt of the mob.
……(repeat to 100K)……

*Without penalty*

……(prompt)……
of biography.

Among the several nations of
modern Europe, France stands in
a class by herself. She is the great
mother of ideas, the molder of the
thoughts of men, the very
fountain of civilization. France is,
or should be, the highest type of
culture, the seat of the
intellectual, the artistic, the
literary, the scientific, and
philosophical life of the world.
……(about 50K words)……
When Napoleon was a boy, he
was a friend of the French people,
and of the Corsicans. Though
born a Catholic, he became a
protestant, and for some time he
lived in the neighborhood of the
French protestants.
……
When he was a boy, he was a
friend of the French people, and
of the Corsicans. .......He was also
a friend of Rousseau, and his first
publication was a poem on the
verses of the great Swiss
philosopher.

*Penalty = 1·15*

*Figure 6.* Case Study on `YaRN-LLaMA2-7b-128k`. Left: fragments of generated text without Contextual Penalty. Right: fragments of generated text with Contextual Penalty. The blue text is repetition part.

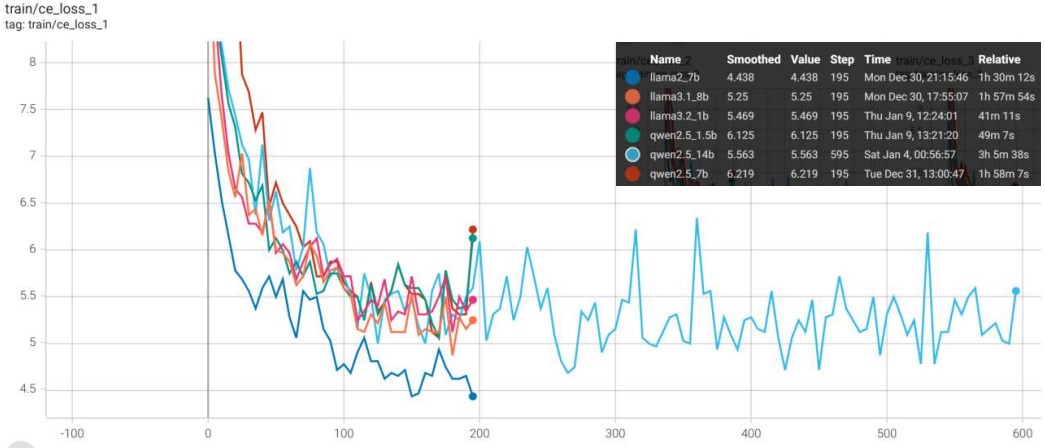

(a) Cross Entropy Loss Training Curve of the First Linear Layer

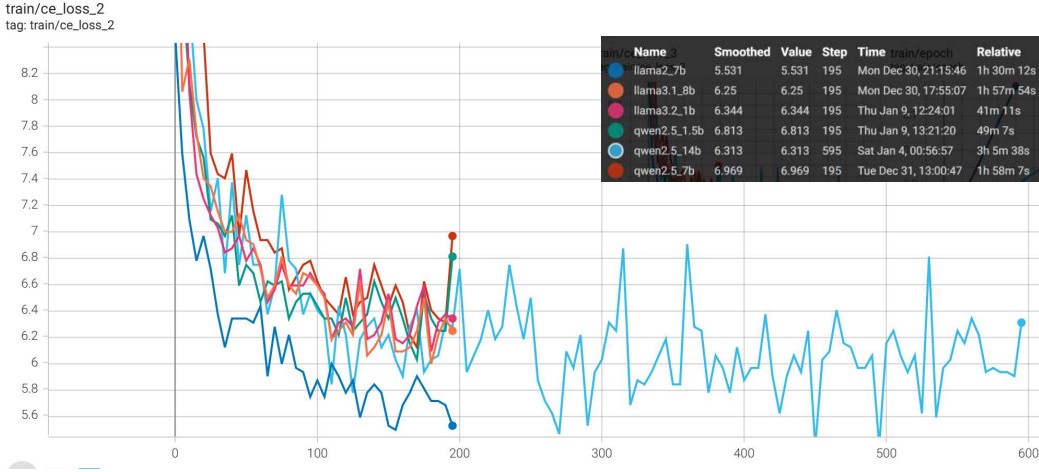

(b) Cross Entropy Loss Training Curve of the Second Linear Layer

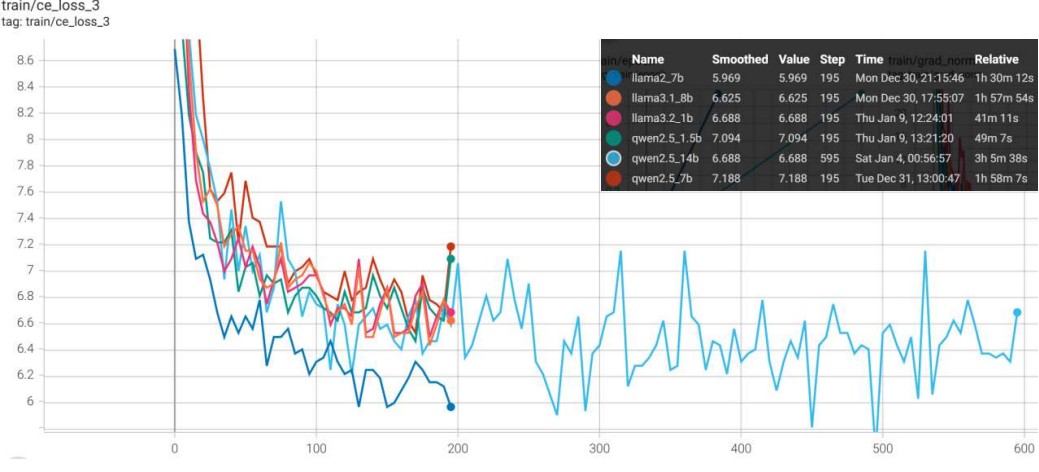

(c) Cross Entropy Loss Training Curve of the Third Linear Layer

*Figure 7.* Cross Entropy Loss Training Curve of Linear Layers

