# OpenReview forum: "TokenSwift: Lossless Acceleration of Ultra Long Sequence Generation"
_ICML.cc/2025/Conference — ICML 2025 poster_

### Official Review · Reviewer_v64u · 2025-03-12

**Overall Recommendation:** 3

**Summary:**

This work proposes a three-pronged approach (TokenSwift) to improving the latency and quality of generating ultra-long sequences: 1) Multi-token prediction similar to Medusa; 2) Using a sparse KV-cache whose elements are selected based on their attention scores (query/key inner product before softmax); 3) token reutilization via n-gram look-ups, somewhat similar to prompt lookup but applied to the generated outputs. The authors show that these techniques significantly improve generation latency compared to selected speculative decoding methods and, when paired with a contextual penalty for repeated outputs, improve the distinct-N metric (unique n-grams / total words) of the output sequences.

**Claims And Evidence:**

* Overall, I believe the empirical evidence supports the author's claims that TokenSwift improves output generation latency for long sequences.
* However, I have concerns regarding some specific claims made otherwise:
    * TokenSwift doesn’t directly address the growing KV cache problem for long sequences. Although the authors note that the baseline methods “would inevitably encounter failures due to KV cache budget constraints” their proposed method would also fail in such a setting as the verification step requires the *full* KV cache. Therefore, in cases where the “growing size of the KV cache would far exceed the allocated length budget”, TokenSwift will also fail. I believe it would be better to reformulate this claim as a benefit with respect to latency instead of claiming other speculative decoding methods would “fail”.
    * The authors claim that TokenSwift enhances diversity of the generated output and also significantly reduces latency. While these claims separately appear to be accurate based on the empirical measures provided, I am concerned TokenSwift seems to inherently *benefit* from repeated / duplicated n-grams in long outputs due to the proposed token reutilization scheme. The reported acceptance rate (AR) for TokenSwift without token reutilization falls significantly as noted in Figure 3. I worry that if sampling was improved such that repetitive content generation was not as prominent, TokenSwift’s performance would be greatly diminished and may no longer offer significant latency improvements over TriForce and other speculative decoding methods. As such, the proposed token utilization is at odds with the eventual desire to have more diverse long generated outputs.

**Essential References Not Discussed:**

None noted.

**Experimental Designs Or Analyses:**

The described experimental design and analyses appear to be sound.

**Methods And Evaluation Criteria:**

* Overall the methods and evaluation criteria for assessing the quality of the speculative decoding process are standard and make sense.
* For assessing the output quality, I find distinct-n to be a useful but insufficient metric on its own. Distinct-n only captures lexical but not semantic diversity. Other text diversity metrics that also capture semantic similarities should be considered. For example: self-BLEU [1], BERTScore [2], or MAUVE [3]. Since repeated content is acknowledged by the authors to also be present in TokenSwift outputs, it would improve the work to further quantify this with additional metrics.

[1] Y. Zhu et al., “Texygen: A Benchmarking Platform for Text Generation Models,” Feb. 06, 2018, arXiv: arXiv:1802.01886. doi: 10.48550/arXiv.1802.01886.
[2] T. Zhang, V. Kishore, F. Wu, K. Q. Weinberger, and Y. Artzi, “BERTScore: Evaluating Text Generation with BERT,” Feb. 24, 2020, arXiv: arXiv:1904.09675. doi: 10.48550/arXiv.1904.09675.
[3] K. Pillutla et al., “MAUVE: Measuring the Gap Between Neural Text and Human Text using Divergence Frontiers,” Nov. 23, 2021, arXiv: arXiv:2102.01454. doi: 10.48550/arXiv.2102.01454.

**Other Comments Or Suggestions:**

* Table 4 Delta T is the number of minutes saved, not hours.
* L433: Text in parentheses states that without a penalty theta = 0 which directly contradicts Table 8 caption.

**Other Strengths And Weaknesses:**

## Strengths:
* Due to the increasing importance of synthetic data and in particular obtaining reasoning traces, this method is of importance to improve the efficiency of such generations.
* This is the first work that I am aware of to apply speculative decoding to long-output generation settings.
* The extensive ablation studies help the reader understand the pros/cons of the proposed method.

## Weaknesses
* The TriForce comparison requires a custom pretrained draft model for the first tier of drafters. Originally, TriForce used a 68M model for this tier whereas in this work we have a 250M model. The large discrepancy between results may influence the overall conclusions reached.
* Some details are not made clear. For example, Table 1 is not referenced in the work. Why are the acceleration rates noted here so much worse than the reference methods? Eg., TriForce achieves acceleration over 2x on A100s with a prompt length of 122K and generation length of 256. Without more details regarding the Table 1 results it’s impossible to compare with previously published values.
* Some unclear choices made in hyperparameter selections for the penalty value.

**Questions For Authors:**

1. How does the generated outputs with and without penalty compare using other non-lexical based metrics such as self-BLEU, BERTScore, or MAUVE? Does the contextual penalty also improve repeated semantic similarity?
2. The authors claim that Triforce and MagicDec are limited to generating 256 or 64 output tokens, respectively (see L040, second column). However, in practice there is no such limit, these are merely the generation settings used by these respective works when benchmarking their methods. Since both of these baselines operate in the typical draft-then-verify framework of speculative decoding, we can simply continue speculating for additional drafting rounds until a desired output length is reached. This is also made clear by the author's use of these baselines for generating sequences greater than the noted limits. Please clarify this statement.
3. What are the settings used for Table 1? Prompt length, data, etc?
4. Why was the penalty value selected as 1.2 instead of 1.5 where a much higher diversity is observed?

**Relation To Broader Scientific Literature:**

TokenSwift builds on prior methods from speculative decoding literature work such as Medusa and prompt-lookup decoding. While these individual components are not overly novel in themselves, their combination and integration is. Further, I believe tweaks to these approaches included in this work such as having non-independent multi-token generation heads is unique and a valuable contribution to the literature. Overall speculative decoding for long context prompts has been studied in some detail but as far as I know this is the first work to directly apply these techniques to the setting of long output generation.

**Theoretical Claims:**

N/A, no novel proofs provided.

---

> ### Author Rebuttal · Authors · 2025-03-31
>
> We sincerely thank you for your time and insightful feedback, which has helped us improve our manuscript.
> > **Q: Reformulate Claim**
>
> **A:** We appreciate your suggestions to replace "fail" with "benefit" and will revise accordingly.
>
> 1. **Dynamic KV Cache Compression (Challenge II)**
> The limitation of baseline methods is not merely budget constraints but their **single-compression design** for long inputs. For long outputs, however, the KV Cache grows dynamically with generated tokens. In contrast, our method maintains a fixed-size KV Cache regardless of output length (no hard budget), as we continuously compress the growing cache.
>
> 2. **Memory Limits**:
> Like all speculative decoding methods, we preserve a full KV Cache for lossless generation. If this cache exceeds **GPU memory** (not a pre-set budget), generation fails—but this is distinct from baselines’ rigid budget-based failures. Note that our focus is on **draft-phase optimization**.
>
> 3. **Example**:
> For TriForce (budget=4096):
>     * Input=100K, Output=256 → KV Cache=4096+256 (valid).
>     * Input=8192, Output=100K → KV Cache=4096+100K (fails).
>
>     Our method fixes the KV Cache size for any input/output length.
>
> > **Q: Sampling Diversity**
>
> **A:**
> We sincerely appreciate your insightful comments regarding the diversity-speedup balance.
>
> 1. **Trade-off**
>    In **Table 6**, our experiments demonstrate that the proposed penalty mechanism enables effective balancing of these two objectives:
>    - Without penalty: Achieves 3.58× speedup with baseline diversity
>    - With θ=1.2 penalty: Maintains 3.27× speedup (only 0.31x reduction) while significantly improving diversity metrics
>
>    This evidences that our approach attains competitive acceleration without severely compromising diversity.
>
> 2. **Main Results**
>    All experiments in **Tables 3-4** already employ θ>1.0 penalty configuration (**Table 10**). The absence of severe repetition issues in these results further confirms the practical viability of our trade-off strategy.
>
> 3. **Self-BLEU**
>     Self-BLEU provides direct measurement:
>      |Metric|Self-BLEU-2|Self-BLEU-3|Self-BLEU-4|Self-BLEU-5|
>      |-|-|-|-|-|
>      |w/o Penalty|0.7899|0.7445|0.7162|0.6966|
>      |w/ Penalty|0.5588|0.4745|0.4115|0.3788|
>
>      Consistent 23.11-31.78% relative reduction across n-grams confirms diversity improvement, aligning with Distinct-N trend(Table 8). BERTScore/MAUVE are inapplicable for our **reference-free open-generation** setting, as they require parallel references.
>
> 4. **Implementation**
>    The adjustable penalty coefficient allows practitioners to prioritize either speed or diversity based on application needs, demonstrating our framework's adaptability.
>
> > **Q: 250M vs. 68M**
>
> **A:**
> For a fair comparison, we used 68M for llama2 and 250M for llama3.1 in the TriForce experiment in Table 3, and the parameter settings of the two draft models are exactly the same. Because the **vocabulary size** of llama3.1 is much larger than that of llama2, the model size becomes larger.Finally, we have open-sourced this model to support future research.
>
> > **Q: Table 1**
>
> **A:** While Table 1 is briefly referenced in **L49–50**, we will explicitly add this context in the revised manuscript to ensure transparency.
>
> We reproduced Triforce's results on A100-80G and PG-19 using identical hyperparameters. The only difference (as noted in L49-50) is that their original experiments were conducted on llama2(**MHA**), while ours used llama3.1(**GQA**).
>
> The observed performance gap arises because **MHA typically requires a KV cache several times larger than GQA**, enabling TriForce’s acceleration.
>
> > **Q: Clarify Statement of Baseline**
>
> **A:** Thank you for your question.
> 1. **Limitations in Prior Work**:
> The values 256 and 64 in Table 1 are just examples. We also optimized TriForce and experimented with generating 100K lengths (**Table 3**). It does achieve a 2x speedup on MHA, but not on GQA.
>
> 2. **Length Restrictions in Baselines**:
> As observed in practice, baselines enforce strict limits on generation length: TriForce caps generation at the KV Cache budget size:
>     ```
>     def update_graph_cache(self, kv_cache=None):
>         self.value_cache[:,:,self.max_budget-(kv_cache.seq_len-self.prefill):self.max_budget] = kv_cache.value_cache[:,:, self.prefill:kv_cache.seq_len].clone()
>         self.key_cache[:,:,self.max_budget-(kv_cache.seq_len-self.prefill):self.max_budget] = kv_cache.key_cache[:,:, self.prefill:kv_cache.seq_len].clone()
>     ```
>
> > **Q: Penalty Value**
>
> **A:** During experiments, we observed that while higher θ (e.g., 1.5) increases diversity, it often leads to incoherent outputs or even garbled text in practice. (Table 8) Thus, we adopted θ=1.2 as a empirically stable default, **balancing diversity and quality**.
>
> > **Q: Typos**
>
> **A:** These were typos in our manuscript, and we will correct them in the revised version. To clarify: when θ=1.0, no repetition occurs.
>
> We hope our answers have resolved your concerns.

---

> > ### Comment · Reviewer_v64u · 2025-04-04
> >
> > I thank the authors for their detailed rebuttal.
> >
> > RE: Single-compression of baseline methods: My understanding of TriForce’ approach is that the tier 1 drafter uses streaming attention from StreamingLLM which does not in principle require that the KV cache grows as context or output length grows. Instead tokens would be evicted as they fall out of the sliding window. Based on my understanding of the “dynamic KV cache” update strategy for TriForce that you implemented for this work I believe the KV cache for TriForce* should not grow with output length. Can you confirm my understanding? For TriForce’s tier-2 retrieval based drafter, I agree that the KV cache grows as output length increases; however, since this is the *same* cache being used by the target model for verification, I disagree that this provides any limit on output length generation other than that provided by the overall memory limit of the hardware used. In this respect, both TriForce and TokenSwift are limited in output length fundamentally by the target model KV cache, not the draft models. The StreamingLLM authors specifically note that they use a “rolling cache at each decoding phase” and if TriForce differs from this implementation it’s a relatively straightforward update to the KV cache update strategy during decoding as you have done here with TriForce*.
> >
> > Overall, I find the claims that TokenSwift enables larger generation outputs than TriForce under a fixed memory budget to be convincing with respect to the original TriForce implementation using a static KV cache update. However, as it only required a small change to TriForce’s KV update strategy to enable the output lengths that you tested with in Table 3, I think the fundamental and more interesting comparison is between TokenSwift and TriForce* which both offer long output generations only limited by the target model’s KV cache.
> >
> > Regarding diversity of outputs, I agree that your repetition penalty is effective in improving sampling diversity and the additional self-BLEU scores highlight this. My main concern with the token reutilization approach is that it inherently benefits from lower diversity of outputs. Therefore, as generation output diversity is improved with future models, TokenSwift’s AR will decay to that of k=0 as noted in Fig 3.
> >
> > Thank you for the note regarding TriForce tier-1 drafter size, this addresses my concerns. I suggest making this explicit in the camera-ready version.
> >
> > Your discussion and clarification of Table 1 is appreciated and makes sense after highlighting GQA vs MHA.
> >
> > Overall and after considering your rebuttal, I believe TokenSwift does offer a practical approach for the current generation of LLMs in which token reutilization offers major benefits. Fundamentally, I remain concerned that token reutilization is at odds with the desire to improve output diversity but while such phenomena exists we can exploit it for efficiency gains as the authors have done here. Based on this, I have raised my score to 3.

---

> > > ### Author Response · Authors · 2025-04-05
> > >
> > > We sincerely appreciate your detailed feedback and recognition of our work. Below we provide point-by-point clarifications to address your concerns:
> > >
> > > **1. Fundamental Differences Between TriForce and TokenSwift**
> > >
> > > The key distinction lies in TriForce's **three-layer** architecture, where the second layer validates the first layer and serves as a draft for the third layer (the final KV-verified model). Specifically:
> > > - TriForce's first layer (68M) employs StreamingLLM-style eviction.
> > > - Its second layer (Retrieval Cache) uses H2O-like compression but only performs a single compression for long prefixes. Crucially, TriForce does not optimize for eviction in long-output scenarios.
> > >
> > > Our core claim is that **TokenSwift's draft phase remains unaffected by KV cache growth**, whereas TriForce's second-layer drafting inevitably suffers performance degradation as KV cache expands (due to unoptimized eviction mechanisms).
> > >
> > > **2. Comparison with TriForce***
> > >
> > > Your observation about the TokenSwift vs. TriForce* comparison is insightful. We clarify that TriForce* represents our optimized implementation of TriForce for fair comparison. The critical insight is: To achieve acceleration in **long-output scenarios**, static KV updates (as in original TriForce) prove insufficient—dynamic cache updates become essential. This fundamental limitation of TriForce's architecture motivates our design philosophy.
> > >
> > > **3. Token Reuse & Diversity**
> > >
> > > We emphasize that "diversity" in our context refers to preventing **meaningless repetitions** (e.g., redundant phrases), not requiring all 100K tokens to be unique. Common lexical repetitions (e.g., "i am", "that is") remain natural and unavoidable.
> > >
> > > Our approach remains effective in long-context generation because:
> > >
> > > a) **Inherent Repetition Necessity**: Any extended text (e.g., novels) naturally contains frequent reuse of common tokens.
> > >
> > > b) **Vocabulary Constraints**: The limited LLM vocabulary (especially high-frequency tokens) guarantees non-degenerate cases (k>0) even in 100K-token generations.
> > >
> > > c) **Practical Motivation**: This linguistic reality directly informs our design—exploiting predictable token recurrence patterns without compromising output quality.
> > >
> > > ---
> > >
> > > We thank you again for your constructive feedback. We would be delighted to provide any clarification or extended analysis required.

---

### Official Review · Reviewer_7LKx · 2025-03-14

**Overall Recommendation:** 3

**Summary:**

As LLMs become bigger in terms of number of parameters, model inference has become computationally expensive leading need for faster and computationally efficient sequence generation. The authors propose the TOKENSWIFT, a new framework to accelerate autoregressive generation for LLMs. TOKENSWIFT utilizes multi token generation to draft multiple tokens in a single forward pass and dynamically updates the KV cache across iterations to achieve lossless acceleration for autoregressive generation.

**Claims And Evidence:**

TOKENSWIFT claims to be the first to accelerate lossless autoregressive generation for LLMs for upto 100k tokens and attain upto 3x speedup across various model architectures. The authors provide empirical evaluation results to substantiate their claims in Table 3 and 4.


Nitpick: The title *"FROM Hours to Minutes"* might be slightly misleading, TOKENSWIFT provides upto 3x speedup and Figure 1 itself showcases sequence generation time coming down from ~5 hours to 1.5 hours.

**Essential References Not Discussed:**

N/A

**Experimental Designs Or Analyses:**

Yes, the authors compare TOKENSWIFT to existing SOTA literature for various models, attention architectures, varying prefill lengths. Thorough ablation studies are conducted to evaluate the pros and cons of different components of the TOKENSWIFT framework.

**Methods And Evaluation Criteria:**

TOKENSWIFT compares autoregressive generation performance across different models and attention architectures (MHA and GQA) and compute latency based with varying prefill lengths. TOKENSWIFT is compared against existing SOTA literature such as Medusa and Triforce for check for performance gains.

**Other Comments Or Suggestions:**

N/A

**Other Strengths And Weaknesses:**

N/A

**Questions For Authors:**

N/A

**Relation To Broader Scientific Literature:**

The work presents novel method for accelerating autoregressive language generation for LLMs. The wider community would benefit from the findings presented to accelerate sequence generation for longer sequences in a lossless manner.

**Theoretical Claims:**

Section 3 discusses the TOKENSWIFT framework which includes multi token generation using additional tuned linear layers, token reutilization and dynamic KV cache management which is based on ranking importance scores suggested in equation 2, with parallel verification of the draft tokens.

---

> ### Author Rebuttal · Authors · 2025-03-28
>
> Thank you very much for your valuable suggestions and acknowledgment of our method. We are encouraged by your recognition of ``our thorough ablation studies`` and ``the practical impact of our method, which achieves up to 3× speedup across diverse architectures``. We also appreciate your note on ``the wider community would benefit from our findings``. Below, we will address your concerns:
>
> > **Q1:** Nitpick: The title "FROM Hours to Minutes" might be slightly misleading
>
> **A1:** We appreciate this observation. We will remove this phrase from the title in the revised version.
>
> We hope our answers have resolved your concerns. If you have any other concerns, please feel free to let us know. Thanks again for your review.

---

### Official Review · Reviewer_zJ1d · 2025-03-15

**Overall Recommendation:** 4

**Summary:**

The paper presents TOKENSWIFT, a framework to accelerate ultra-long sequence generation (up to 100K tokens) in large language models (LLMs) with lossless accuracy, addressing the time-intensive nature of such tasks (e.g., LLaMA3.1-8B taking 5 hours). It tackles three challenges—frequent model reloading, dynamic KV cache management, and repetitive content—via multi-token generation with token reutilization, dynamic KV cache updates, and a contextual penalty mechanism. Main findings include a 3x speedup across models (1.5B-14B) and architectures (MHA, GQA), reducing LLaMA3.1-8B’s 100K-token generation to 90 minutes, with improved diversity (Distinct-n) and superiority over baselines like TriForce* and Medusa*, as validated on PG-19.

**Claims And Evidence:**

The claims are well-supported by evidence. The 3x speedup is convincingly demonstrated through experiments (Tables 3, 4), with detailed comparisons to AR and baselines, and time savings (e.g., 3.5 hours for Qwen2.5-14B) adding clarity. Lossless acceleration is theoretically backed by speculative decoding (Appendix A) and empirically implied by matching AR outputs. The “first for 100K-token lossless acceleration” claim aligns with cited limitations of prior work (e.g., TriForce’s 256-token limit, Table 1), though “minimal training cost” lacks detailed quantification beyond training three linear layers (§3.1), slightly weakening its evidential strength.

**Essential References Not Discussed:**

N/A

**Experimental Designs Or Analyses:**

The experimental design is sound, testing TOKENSWIFT across diverse models (e.g., LLaMA3.1-8B, Qwen2.5 series) and lengths (20K-100K tokens) on a single A100 GPU (§4.1), with results averaged over five runs to reduce randomness (Table 3). Ablations on sampling methods (Table 12), temperature (Table 13), and penalty windows (Table 15) are rigorous, validating robustness. However, the resource cost of maintaining full KV cache for verification isn’t quantified, which could affect scalability claims on resource-constrained settings—an area for minor clarification.

**Methods And Evaluation Criteria:**

The proposed methods—multi-token generation, dynamic KV updates, and contextual penalties—are sensible for ultra-long sequence generation, directly addressing identified bottlenecks. The evaluation criteria, including speedup (latency ratio), acceptance rate
$\alpha$, and Distinct-n for diversity, are appropriate and standard for assessing acceleration and quality. Using PG-19 as a benchmark is reasonable for long-sequence tasks, though its representativeness for all LLM applications could be broader.

**Other Comments Or Suggestions:**

N/A

**Other Strengths And Weaknesses:**

N/A

**Questions For Authors:**

N/A

**Relation To Broader Scientific Literature:**

N/A

**Theoretical Claims:**

I reviewed the primary theoretical claim of lossless acceleration, supported by the proof in Appendix A, which demonstrates that TOKENSWIFT’s speculative decoding (SD) output distribution $p_{SD} $ equals the target model’s distribution $q_{target} $.

---

> ### Author Rebuttal · Authors · 2025-03-31
>
> We sincerely thank you for your time and constructive feedback. We are encouraged by your positive assessment that ``our experiments convincingly demonstrate a 3× speedup``, ``the sensible design of our method for ultra-long sequence generation``, and ``the rigorous ablation studies on sampling methods, temperature, and penalty windows``. Below, we address their concerns point-by-point.
>
> >**Q1:** though “minimal training cost” lacks detailed quantification beyond training three linear layers (§3.1), slightly weakening its evidential strength.
>
> **A1:** Thank you for highlighting this point. We will provide a detailed breakdown of the quantized training cost in the revised manuscript. Specifically, for any large language model, our approach requires only ``γ（Number of tokens generated in parallel） × hidden_size × hidden_size`` trainable parameters. The specific training time is shown in **Appendix I**. For example, the training time for models 8B does not exceed two hours. The concrete results for the LLMs used in our experiments are summarized in the table below:
>
> | Model | Llama3.1-8B | Llama2-7B | Qwen2.5-1.5B | Qwen2.5-7B | Qwen2.5-14B |
> | -------- | -------- | -------- | -------- | -------- | -------- |
> | Param. | 50.3M | 50.3M | 7.1M | 38.5M | 78.6M |
>
> > **Q2:** However, the resource cost of maintaining full KV cache for verification isn’t quantified, which could affect scalability claims on resource-constrained settings—an area for minor clarification.
>
> **A2:** Thank you for your suggestion. The full KV Cache memory overhead can indeed be derived from the model configuration. Taking Llama3.1-8B (GQA) as an example (batch size=1, bfloat16), the **peak** memory usage is calculated as:
> 2 × Layer Num × Batch Size × Seq Len × KV Heads × Head Dim × Bytes
> = 2 × 32 × 1 × 102,400 × 8 × 128 × 2 bytes ≈ 13.4 GB.
> The table below summarizes the **peak** memory requirements for other LLMs in our study when maintaining full KV Cache:
>
> | Model | Llama3.1-8B | Llama2-7B(MHA) | Qwen2.5-1.5B | Qwen2.5-7B | Qwen2.5-14B |
> | -------- | -------- | -------- | -------- | -------- | -------- |
> | Param. | 13.4G | 53.7G | 2.9G | 5.9G | 20.1G |
>
> We hope the above response can resolve your questions and concerns. Please let us know if there is any further question! Thanks again for your review.

---

> > ### Comment · Reviewer_zJ1d · 2025-04-05
> >
> > Thank authors for the updated experiments and they have addressed my questions and concerns. I will keep my original positive score.

---

> > > ### Author Response · Authors · 2025-04-05
> > >
> > > We sincerely appreciate your insightful feedback and dedicated efforts in reviewing our work. Thank you for recognizing the improvements and maintaining your positive evaluation of our paper.

---

### Decision · Program_Chairs · 2025-05-01

**Decision:**

Accept (poster)

**Comment:**

The paper introduces TokenSwift, tackling three latency problems in autoregressive decoding  by combining multi-token generation, dynamic KV cache updates, and token reutilization with contextual penalties. The approach is empirically solid, with a 3× speedup validated across models and lengths without sacrificing output quality, and theoretically justified for lossless acceleration. While some concerns were raised about diversity tradeoffs and baseline comparisons, the authors responded thoroughly, and reviewers generally agreed on the method's merit and practical significance. I recommend acceptance.